# Decoding a neural circuit controlling global animal state in *C. elegans*

Patrick Laurent[1†], Zoltan Soltesz[1†], Geoffrey M Nelson[2], Changchun Chen[1], Fausto Arellano-Carbajal[1,3], Emmanuel Levy[1,4], Mario de Bono[1*]

[1]Laboratory of Molecular Biology, Cambridge, United Kingdom; [2]Cell Biology Division, MRC Laboratory of Molecular Biology, Cambridge, United Kingdom; [3]School of Natural Science, Universidad Autonoma de Queretaro, Santiago de Querétaro, Mexico; [4]Weizmann Institute of Science, Rehovot, Israel

**Abstract** Brains organize behavior and physiology to optimize the response to threats or opportunities. We dissect how 21% $O_2$, an indicator of surface exposure, reprograms *C. elegans*' global state, inducing sustained locomotory arousal and altering expression of neuropeptides, metabolic enzymes, and other non-neural genes. The URX $O_2$-sensing neurons drive arousal at 21% $O_2$ by tonically activating the RMG interneurons. Stimulating RMG is sufficient to switch behavioral state. Ablating the ASH, ADL, or ASK sensory neurons connected to RMG by gap junctions does not disrupt arousal. However, disrupting cation currents in these neurons curtails RMG neurosecretion and arousal. RMG signals high $O_2$ by peptidergic secretion. Neuropeptide reporters reveal neural circuit state, as neurosecretion stimulates neuropeptide expression. Neural imaging in unrestrained animals shows that URX and RMG encode $O_2$ concentration rather than behavior, while the activity of downstream interneurons such as AVB and AIY reflect both $O_2$ levels and the behavior being executed.

*For correspondence: debono@ mrc-lmb.cam.ac.uk

†These authors contributed equally to this work

## Introduction

Mammals adopt different global states in response to threats or opportunities by coordinated changes in physiology and neurochemistry that optimize and focus their response to the situation at hand (*LeDoux, 2012*). Hallmarks of such global states include arousal, a reconfiguring of the relative importance given to different sensory cues, and altered physiology due to endocrine feedback (*LeDoux, 2012*). Examples of global states include those evoked by a potential mate (*Pfaff et al., 2008*), by predators (*Martinez et al., 2008*; *Motta et al., 2009*), or by nutritional state (*Atasoy et al., 2012*; *Sternson et al., 2013*). Understanding how global states are encoded in neural circuits is of interest because they may provide insights into subjective behaviors: for example, fear, aggression, and hunger (*LeDoux, 2012*; *Damasio and Carvalho, 2013*).

The exact nature of global organismic states is poorly understood. A switch in global state is thought to involve recruitment of many brain circuits whose individual activities are dynamically assembled to address the circumstances faced by the animal. A powerful entry point to study circuits orchestrating such states has involved identifying small populations of neurons whose activation or inhibition can evoke features of the states. For example, in mouse, optogenetic activation of neurons in the ventromedial hypothalamus can induce aggressive behavior (*Lin et al., 2011*), while optogenetic control of basolateral terminals in the amygdala's central nucleus can regulate anxiety-like states (*Tye et al., 2011*). How these small populations of neurons modulate different areas of the brain, how their functional effects depend on the state of other circuits, and how their activity is itself controlled, are open questions.

Invertebrates also adopt different behavioral states, for example, in response to potential mates (*Villella and Hall, 2008*), predators (*Henschel, 1990*; *Uma and Weiss, 2012*), conspecific rivals

**eLife digest** From humans to worms, animals must respond appropriately to environmental challenges to survive. Starving animals must conserve energy while they seek food; animals that encounter a predator must fight or flee. These responses involve the animals re-programming their bodies and behavior, and, in humans, are thought to coincide with feelings or emotions such as 'hunger' and 'fear'. Understanding these states in humans is difficult, but studies of simpler animals may provide some insights.

The microscopic worm *Caenorhabditis elegans* offers a unique advantage to these studies because it has the most precisely described nervous system of any animal. The worm lives in rotting fruit, but it avoids the fruit's surface, perhaps because there is an increased risk of it drying out or being eaten by predators. Microbes that grow within the rotting fruit reduce the oxygen level below the 21% oxygen found in the surrounding air, and so one strategy that *C. elegans* uses to avoid surface exposure is to continuously monitor the oxygen concentration. If the worm senses that the oxygen level is approaching 21%, which suggests it is nearing the surface, it reverses and turns around. If it cannot find a lower-oxygen environment, the worm switches to continuous rapid movement until it locates such an environment, and adapts its body for surface exposure.

Laurent, Soltesz et al. sought to understand the circuit of neurons that controls this switch. Monitoring gene expression in the worms revealed that specific oxygen-sensing neurons help generate the widespread changes that occur in the worm's body. These neurons also control the switch in the worm's behavior. Sensory neurons relay signals to downstream neurons that act on muscles to alter behavior. Neurons typically communicate with other neurons via specific connections; but neurons can also release signaling molecules, which act like 'wireless' signals and can affect many other cells. Laurent, Soltesz et al. showed that both kinds of signaling are needed to change the worm's behavior, and suggest that the release of signaling molecules may explain the widespread effects of 21% oxygen on the worm.

Laurent, Soltesz et al. then monitored the activity of neurons in freely moving worms, and found that some neurons appear to encode and relay specific sensory information. Other neurons encode the behavior the animal is performing, and yet others can encode both kinds of information. To confirm which neurons control particular behavioral responses, Laurent, Soltesz et al. measured changes in the worm's behavior after destroying or altering specific cells, or while they used light-based techniques to artificially excite or inhibit specific neurons.

At a simple level the worm's response to 21% oxygen resembles the response of a mammal to a dangerous environment: both become more aroused, change how they respond to other sensory cues, and adapt both their bodies and behavior. As such, *C. elegans* provides a great model to explore at a small and accessible scale how changes in animals' states are generated.

(*Dow and von Schilcher, 1975*), and nutritional state (*Gaudry and Kristan, 2012*). Some of these responses share features of global organismic states in mammals, and may provide insights into how the latter are encoded. A complication in understanding how circuits encode global animal states is that the cues evoking them are often complex. For example, a male *Drosophila* courting a female fly responds to visual, olfactory, and gustatory cues (*Villella and Hall, 2008*); in *Caenorhabditis elegans*, starvation resets many sensory responses, including gustation (*Saeki et al., 2001*), olfaction (*Tsunozaki et al., 2008*), and thermotaxis (*Hedgecock and Russell, 1975*), but how starvation is sensed and communicated to sensory circuits is poorly understood (*Milward et al., 2011*). A way around this problem is to identify animal state changes that are robustly linked to single identified sensory inputs, providing a defined entry point to dissect how neural networks are reconfigured to encode global states (e.g., *Chamero et al., 2007*; *Kubli and Bopp, 2012*; *Dewan et al., 2013*).

Surface exposure is hazardous for some terrestrial invertebrates, for example, due to desiccation and predation, and triggers a switch in behavioral state as animals seek to escape it. Oxygen concentration $[O_2]$ is 21% at the surface but lower in buried spaces due to biomass respiration. The nematode *C. elegans* can recognize surface exposure by measuring $[O_2]$ (*Gray et al., 2004*; *Persson et al., 2009*). An increase in $O_2$ from 19% to 21% elicits avoidance behaviors: animals reverse and

change their direction of travel (*McGrath et al., 2009*; *Persson et al., 2009*). If after these maneuvers *C. elegans* fails to find an environment with lower [$O_2$], they become highly active, suggesting a simple form of arousal associated with escape behavior (*Busch et al., 2012*). This activated state is sustained for at least 2 hr, and potentially until animals locate an environment with lower $O_2$ levels (*Busch et al., 2012*).

Avoidance and escape from 21% $O_2$ are driven principally by $O_2$-sensing neurons called URX, AQR, and PQR (*Gray et al., 2004*; *Cheung et al., 2005*; *Zimmer et al., 2009*; *Busch et al., 2012*; *Couto et al., 2013*). The pair of URX neurons is the most important, and is necessary and sufficient to mediate these responses. In these neurons, rising $O_2$ stimulates an atypical $O_2$-binding soluble guanylate cyclase composed of GCY-35 and GCY-36 subunits (*gcy*, *guanylate cyclase*), leading to cGMP channel opening, and activation of L-type $Ca^{2+}$ channels (*Cheung et al., 2004*; *Gray et al., 2004*; *Zimmer et al., 2009*; *Busch et al., 2012*; *Couto et al., 2013*). The response of URX neurons (as well as AQR and PQR) to 21% $O_2$ is tonic, that is, non-adapting, and alters other behaviors besides promoting rapid movement. At 19% $O_2$, *C. elegans* strongly avoids carbon dioxide ($CO_2$), but at 21% $O_2$ tonic URX signaling suppresses this avoidance (*Carrillo et al., 2013*; *Kodama-Namba et al., 2013*). Tonic signaling from the $O_2$ sensors induces *C. elegans* to leave depleting food patches at much higher rates in 21% $O_2$ (*Lima and Dill, 1990*; *Milward et al., 2011*). This behavior is consistent with ecological studies of many species showing that animals tend to leave foraging sites when threat levels rise (*Lima and Dill, 1990*). The $O_2$-sensing neurons also promote aggregation and accumulation where bacterial food is thickest (*Coates and de Bono, 2002*; *Gray et al., 2004*; *Rogers et al., 2006*). Besides reconfiguring behavior, $O_2$-sensing neurons alter physiology: they regulate lifespan (*Liu and Cai, 2013*) and body size (*Mok et al., 2011*). In summary, *C. elegans* perceives 21% $O_2$ as a threat, and responds to it via a discrete set of $O_2$-sensing neurons whose tonic activity coordinates an altered global organismic state.

The striking behavioral switch observed in natural *C. elegans* isolates at 21% $O_2$ cannot be studied in the laboratory reference strain, N2 (Bristol), due to a gain-of-function mutation in a FMRFamide-like peptide (FLP) receptor, called NPR-1 (*neuropeptide receptor family*) that arose during domestication (*de Bono and Bargmann, 1998*; *Rockman and Kruglyak, 2009*; *Weber et al., 2010*). Knocking out *npr-1* restores to N2 (Bristol) animals strong responses to 21% $O_2$. NPR-1 is expressed in about 20 neural types, including the $O_2$-sensing neurons AQR, PQR, and URX, and their post-synaptic partners RMG and AUA. The NPR-1 215V receptor alters the function of several of these neurons, including URX and RMG, although the mechanisms involved are unclear (*Coates and de Bono, 2002*; *Macosko et al., 2009*; *Carrillo et al., 2013*; *Kodama-Namba et al., 2013*).

Here, we investigate how $O_2$-sensing neurons evoke a change in global *C. elegans* state in response to 21% $O_2$. We trace information flow from a defined entry point—the URX $O_2$ sensors—to downstream circuits that implement the change in state.

## Results

### $O_2$ levels can reprogram *C. elegans* gene expression through $O_2$-sensing neurons

We hypothesized that tonic signaling from the $O_2$-sensing neurons URX, AQR, and PQR could reprogram gene expression according to ambient $O_2$ levels. To investigate this possibility, we analyzed the transcriptomes of *npr-1(null)* mutants (referred to as *npr-1*) and *gcy-35*; *npr-1* young adult animals grown at 21% and 7% $O_2$ using RNA sequencing (see 'Materials and methods'). URX, AQR, and PQR neurons do not respond to $O_2$ changes in *gcy-35*; *npr-1* mutants, and exhibit activity levels found in *npr-1* animals kept at 7% $O_2$ (*Persson et al., 2009*; *Zimmer et al., 2009*; *Busch et al., 2012*). To prevent *npr-1* animals from aggregating, which induces gene expression changes that would confound our analysis (*Andersen et al., 2014*), we grew animals at low density. We carried out three sets of comparisons. To identify genes whose expression is $O_2$-modulated we compared the transcriptomes of *npr-1* animals grown at 21% and 7% $O_2$ (*Supplementary file 1*). To identify genes regulated by GCY-35-dependent signaling, we compared the transcriptomes of *npr-1* and *gcy-35*; *npr-1* animals grown at 21% $O_2$ (*Supplementary file 2*). And to identify and exclude genes whose expression is altered by growth at different $O_2$ levels independently of *gcy-35*, we compared the transcriptomes of *gcy-35*; *npr-1* animals grown at 21% and 7% $O_2$ (*Supplementary file 3*). We then intersected the three comparisons, using as a cut-off q value of 0.05 (see 'Materials and methods').

If our hypothesis was correct, many of the genes identified as differentially expressed in *Supplementary file 1* (*npr-1* 21% $O_2$ vs *npr-1* 7% $O_2$) should also be identified in *Supplementary file 2* (*npr-1* 21% $O_2$ vs *gcy-35; npr-1* 21% $O_2$) but not in *Supplementary file 3* (*gcy-35; npr-1* 21% $O_2$ vs *gcy-35; npr-1* 7% $O_2$). Consistent with this, we found that 152/210 genes differentially expressed between *npr-1* 7% $O_2$ versus *npr-1* 21% $O_2$ were also found in the *gcy-35; npr-1* 21% $O_2$ versus *npr-1* 21% $O_2$ comparison. Expression of 72/152 of these genes was not significantly altered by $O_2$ experience in *gcy-35; npr-1* animals (*Supplementary file 4*). For almost all of these genes, 71/72, disrupting *gcy-35* had a similar effect on gene expression as growing *npr-1* animals at 7% $O_2$ (*Supplementary file 4*). Genes whose expression was regulated by $O_2$ experience in a GCY-35-dependent way encoded proteins involved in metabolism (e.g., *elo-6*, a polyunsaturated fatty acid elongase; *folt-2*, a folate transporter; *mai-2*, a mitochondrial intrinsic ATPase inhibitor protein), oxidation–reduction reactions (e.g., the cytochrome p450s, *cyp-35C1* and *cyp-34A2*), and proteolysis (e.g., the cathepsins *cpr-4* and *cpr-6*), suggesting broad changes not limited to the nervous system (*Supplementary file 4*).

## URX neurons provide a defined entry point into the circuit controlling a switch in animal state

Efficient escape from aversive environments requires coordinated movement that avoids conflicting behaviors. When exposed to 21% $O_2$ in the presence of food, *npr-1* animals not only increased their speed of movement (*Busch et al., 2012*) (*Figure 1A*), but also increased the persistence of their forward travel by inhibiting spontaneous short reversals (*Figure 1B,C*). Mutations in *gcy-35*, which abolish the $O_2$ responsiveness of URX, AQR, and PQR neurons, disrupted $O_2$ modulation of both speed (*Busch et al., 2012*) and spontaneous reversal frequency (*Figure 1B*). Expressing *gcy-35* selectively in URX neurons restored rapid movement and sustained inhibition of reversals at 21% $O_2$. URX neurons thus provide a defined entry point to the circuit controlling the behavioral state switch evoked by 21% $O_2$ in *npr-1* animals.

## Stimulating RMG inhibits reversals and induces rapid and persistent forward movement

Among the three $O_2$-sensing neurons, URX neurons uniquely make gap junctions and reciprocal synaptic connections with the RMG interneurons (*White et al., 1986*; wormwiring.org; *Figure 1—figure supplement 1*). The RMG interneurons respond to a 7–21% rise in $O_2$ with a sustained increase in $Ca^{2+}$, and ablating RMG disrupts behavioral responses evoked by this $O_2$ switch (*Busch et al., 2012*). To show that $O_2$-evoked $Ca^{2+}$ responses in RMG reflect input from URX, we ablated URX and imaged RMG using the YC2.60 $Ca^{2+}$ sensor (*Horikawa et al., 2010*). When URX was ablated, $O_2$ stimuli failed to evoke $Ca^{2+}$ responses in RMG (*Figure 1D*), and RMG $Ca^{2+}$ resembled that found in *tax-4* mutants, which lack the cGMP-gated channel required for URX, AQR, and PQR neurons to transduce $O_2$ stimuli (*Figure 1D*). These data suggest that $O_2$-evoked $Ca^{2+}$ responses in RMG are driven by URX input.

Is depolarizing RMG sufficient to switch *C. elegans* behavioral state? To address this we used channelrhodopsin-2 (ChR2; *Nagel et al., 2005*) to selectively stimulate RMG in *npr-1* animals kept at 7% $O_2$. Stimulating RMG inhibited spontaneous reversals and induced rapid and persistent forward movement for as long as blue light was on (*Figure 1E,F*). Stimulating RMG neurons is thus sufficient to confer a highly active locomotory state on *npr-1* animals kept at 7% $O_2$.

In *npr-1* animals, URX neurons respond in a graded manner to changes in $O_2$ between 7% and 21%, and evoke graded increases in locomotory rate according to final $O_2$ concentration (*Cheung et al., 2005*; *Busch et al., 2012*). To investigate the dynamic range of URX–RMG signaling, and to examine if graded RMG activity can evoke graded changes in the animal's speed, we stimulated RMG in animals kept at either 7% $O_2$ or 21% $O_2$. The effects of activating RMG using ChR2 summated with input from the $O_2$ sensory circuit: animals kept at 21% reached higher speeds and inhibited reversals more strongly than animals kept at 7% (*Figure 1E,F*). These data suggest that the dynamic range of the circuit permits higher levels of RMG activation to evoke qualitatively similar but quantitatively stronger behavioral responses.

RMG and URX are connected by gap junctions and by reciprocal chemical synapses. Stimulating RMG using ChR2 could therefore alter behavior by activating URX. To test this, we selectively activated RMG using ChR2 in animals ablated for URX (and AQR and PQR). Ablated animals robustly

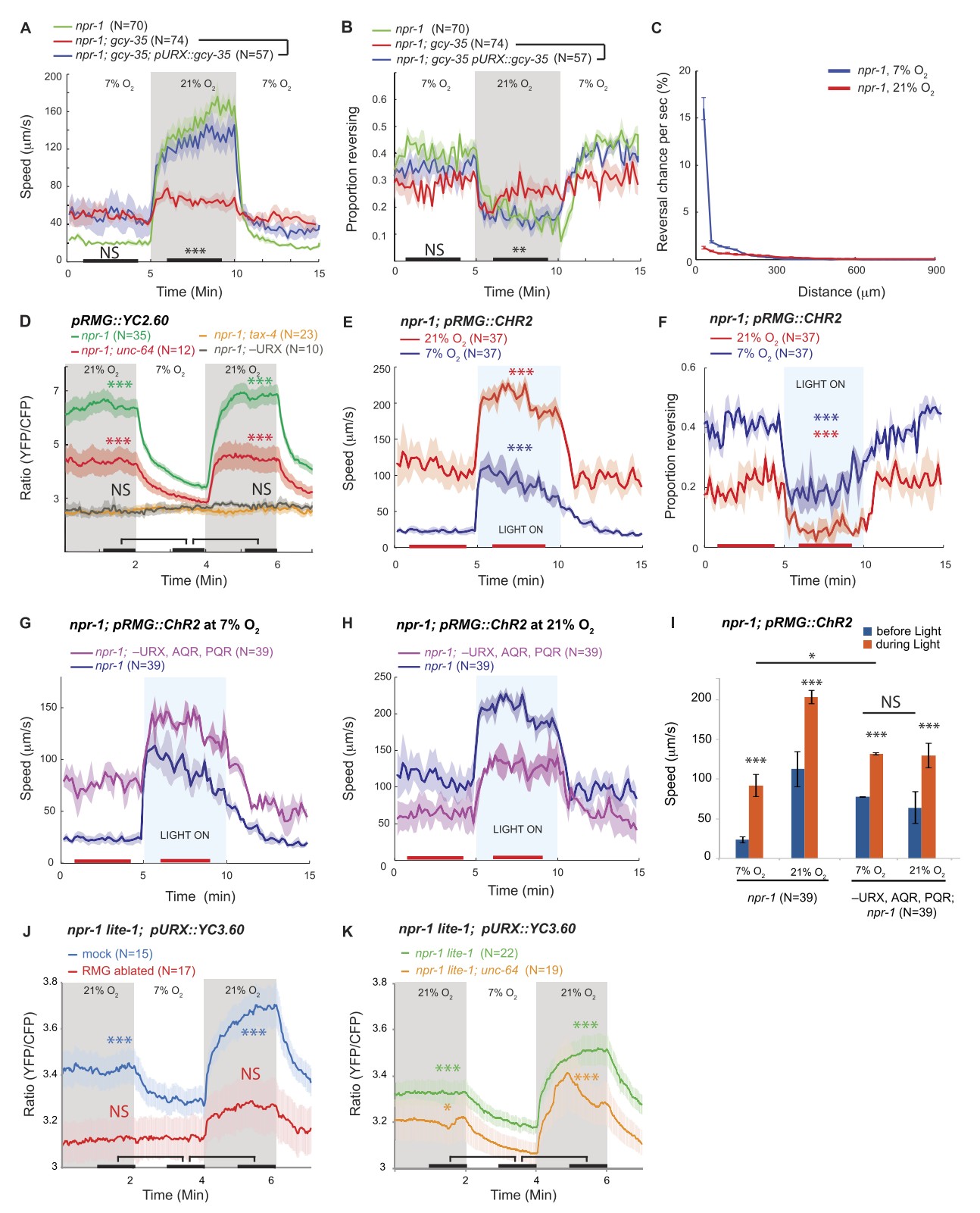

**Figure 1**. RMG activation induces rapid and persistent forward movement. (**A** and **B**) URX O₂ sensors provide an entry point to the circuit controlling response to 21% O₂. Selective expression of GCY-35 in URX neurons restores rapid (**A**) and persistent (**B**) forward movement at 21% O₂ to *gcy-35*; *npr-1* animals on food. Statistics compare rescued (blue) and mutant (red) animals at time points indicated by the black bars. (**C**) 21% O₂ causes *npr-1* animals on

*Figure 1. continued on next page*

*Figure 1. Continued*

food to suppress the short, frequent reversals observed at 7% $O_2$. Reversal probability is calculated per 1 s. (**D**) Ablating URX abolishes the $Ca^{2+}$ responses evoked in RMG by 21% $O_2$; *unc-64* syntaxin loss-of-function mutants show a partial reduction in this response. Here, and in subsequent panels, black bars indicate time intervals for statistical comparison of responses at 21% and 7% $O_2$ using the Mann–Whitney U test. ***p<0.001; **p<0.01; *p<0.05; NS, not significant. (**E** and **F**) Stimulating RMG using channelrhodopsin evokes rapid movement (**E**) and inhibits backward movement (**F**) in *npr-1* animals kept at either 7% or 21% $O_2$ with food. Here and in subsequent panels, red bars indicate time intervals used for statistical comparisons of responses when light is on to when it is off. ***p<0.001, **p<0.01; *p<0.05; NS, not significant. (**G–I**) Channelrhodopsin stimulation of RMG can induce rapid movement when AQR, PQR, and URX neurons are ablated. Data plotted in (**I**) are replotted from **G** and **H**. (**J**) URX $Ca^{2+}$ responses evoked by 21% $O_2$ are strongly attenuated when RMG is ablated. (**K**) URX neurons retain $O_2$-evoked $Ca^{2+}$ responses in *unc-64* syntaxin mutants, although baseline $Ca^{2+}$ is reduced. Each line in this panel (and, unless specified, in subsequent panels) represents the mean response of all animals of one genotype or condition. Error bars (lighter shading) in all panels show standard error of the mean. Gray areas indicate periods of higher $O_2$ concentration; blue areas indicate periods with blue light (0.26 mW/mm²) on; orange areas indicate periods with green light (0.64 mW/mm²) on.

The following figure supplement is available for figure 1:

**Figure supplement 1**. Simplified circuitry associated with the URX, AQR, and PQR neurons dissected in this paper.

increased their speed and suppressed reversals upon light-activation of RMG, both at 7% and at 21% $O_2$ (*Figure 1G,H*). As expected if $O_2$ input was lost, the speed responses of the ablated animals were not influenced by the $O_2$ levels (*Figure 1I*). The ability of RMG stimulation to induce rapid movement and inhibit reversals in the absence of URX, AQR, and PQR neurons suggests that feed-forward signaling from RMG to other neurons evokes these behaviors.

Our results did not exclude that RMG influences URX activity. To investigate this possibility, we ablated RMG neurons in L1 larvae and imaged URX $Ca^{2+}$ responses in young adults 2 days later. *npr-1* animals lacking RMG showed a much smaller URX $Ca^{2+}$ response to a 7–21% $O_2$ stimulus compared to mock ablated controls (*Figure 1J*). Thus, not only does URX activate RMG in response to rising $O_2$, but RMG input somehow regulates URX excitability. Although the gap junctions and synaptic connections between URX and RMG provide a direct route for such communication, we cannot exclude more indirect mechanisms. However, consistent with a role for synaptic input in sustaining URX excitability, partial loss-of-function mutants of *unc-64* syntaxin, which have deficits in synaptic release, showed reduced $Ca^{2+}$ levels in URX and RMG both at 7% and 21% $O_2$ (*Figure 1D,K*). URX neurons could still evoke $Ca^{2+}$ responses in RMG in *unc-64* mutants (*Figure 1D*). However, since the *unc-64* allele we used is a partial loss-of-function (null mutants are dead), we cannot draw firm conclusions about the relative roles of gap junctions and synaptic transmission in mediating URX–RMG communication.

## FLP-21–NPR-1 signaling limits the output of the RMG circuitry

Excessive signaling from a hyperactive neuropeptide receptor, NPR-1 215V, prevents N2 (Bristol) animals from moving rapidly on food at 21% $O_2$ (*de Bono and Bargmann, 1998*; *Cheung et al., 2005*). The RMG neurons are one site of action for this receptor: expressing NPR-1 215V selectively in RMG reduces the locomotory activity of *npr-1* animals (*Macosko et al., 2009*). Does NPR-1 215V expression in RMG inhibit responses evoked by a switch from 7% to 21% $O_2$? *npr-1* animals that expressed NPR-1 215V in RMG did not exhibit long-lasting changes in locomotion in response to changes in $O_2$ (*Figure 2*, *Figure 2—figure supplement 1A,B*). However, they retained the transient bout of reversals and reorientation triggered by a sharp rise in $O_2$ (*Figure 2—figure supplement 1C*). Thus, NPR-1 215V receptor signaling in RMG selectively suppresses the sustained switch in locomotory behavior evoked by high and low $O_2$. A different circuit may act downstream of $O_2$ sensors to evoke transient avoidance responses, consistent with the observation that activating RMG using ChR2 failed to induce reversals.

The FMRFamide neuropeptide encoded by the *flp-21* gene is an in vivo ligand for NPR-1 (*Rogers et al., 2003*). *flp-21* is expressed in several neurons, including RMG itself (*Kim and Li, 2004*; *Macosko et al., 2009*). FLP-21 peptide release could therefore serve to limit the $O_2$-evoked behavioral state switch by activating NPR-1 signaling in RMG. Consistent with this, deleting *flp-21* caused animals expressing the natural *npr-1 215F* or derived *npr-1 215V* allele, but not *npr-1(null)* mutants, to increase the amplitude of their $O_2$-evoked behavioral switch (*Figure 2A–C*).

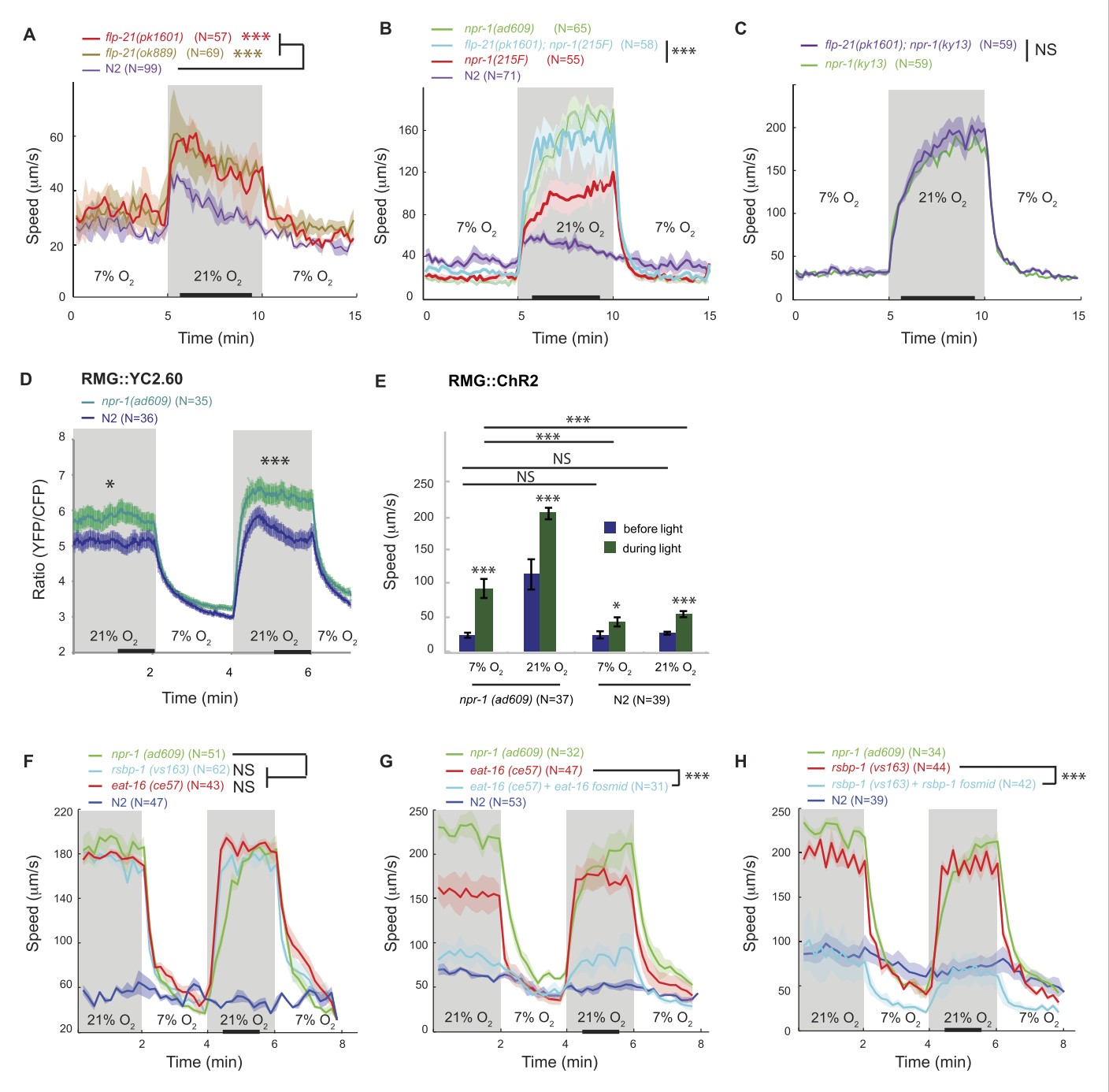

**Figure 2**. FLP-21/NPR-1 ligand/receptor signaling limits RMG circuit output downstream of RMG cell body $Ca^{2+}$. (**A–C**) Disrupting the *flp-21* FMRF-like neuropeptide enables N2 animals to respond to 21% $O_2$ with a persistent increase in locomotory activity (**A**), and increases the amplitude of such responses in animals expressing the natural *npr-1 215F* receptor allele (**B**) but not in *npr-1(ky13null)* mutants (**C**). (**D**) The *npr-1 215V* allele only slightly reduces the $Ca^{2+}$ responses evoked by 21% $O_2$ in RMG. (**E**) RMG stimulation using ChR2 only weakly stimulates movement in N2 animals, both at 7% and 21% $O_2$, contrasting with its effects in *npr-1* animals. (**F–H**) Knocking out *eat-16*, a member of the RGS7 family that inhibits $G_q$ signaling, or its interacting partner *rsbp-1*, enables N2 animals to switch to rapid movement at 21% $O_2$.

The following figure supplements are available for figure 2:

**Figure supplement 1**. *npr-1* GPCR signaling inhibits long-lasting but not transient responses to 21% $O_2$ in feeding animals.

**Figure supplement 2**. Differences in YC2.60 sensor expression may partially account for differences in YFP/CFP ratios in RMG between N2 and *npr-1* animals.

## The NPR-1 neuropeptide receptor acts downstream to $Ca^{2+}$ to limit output of RMG interneurons

How does the NPR-1 neuropeptide receptor inhibit RMG function? Using the YC2.60 sensor, we compared the $Ca^{2+}$ responses evoked in RMG by a 7–21–7% $O_2$ stimulus train in N2 and *npr-1* mutants. N2 animals exhibited reduced $Ca^{2+}$ responses in RMG compared to *npr-1* mutants (*Figure 2D*), which would be consistent with NPR-1 215V inhibiting RMG $Ca^{2+}$ signaling. However, the effects, although statistically significant, were surprisingly small compared to the strong inhibitory effects of NPR-1 215V signaling on $O_2$-evoked behaviors. More importantly, the promoter combination targeting YC2.60 expression to RMG (*pncs-1::cre* and *pflp-21::LoxP-STOP-LoxP:: YC2.60*) drove lower expression in N2 compared to *npr-1* animals (*Figure 2—figure supplement 2A*). Lower sensor expression correlated with lower YFP/CFP ratio at 21% $O_2$ (*Figure 2—figure supplement 2B*), suggesting that sensor expression differences could account for much of the YFP/ CFP ratio differences between N2 and *npr-1* animals (see below for an explanation of why *pflp-21* drove lower RMG expression in N2 compared to *npr-1* animals).

These observations led us to speculate that the main inhibitory role of NPR-1 was downstream of $Ca^{2+}$ influx. To test this, we compared the effects of stimulating RMG neurons with ChR2 in N2 and *npr-1* animals. If NPR-1's inhibitory role is predominantly downstream of $Ca^{2+}$ entry, ChR2-activation of RMG should have little effect in N2 animals. Consistent with this, light activation of RMG triggered only a small increase in N2 locomotory activity, regardless of whether animals were kept at 21% or 7% $O_2$, and in marked contrast to its effects in *npr-1* mutants (*Figure 2E*). Most compellingly, although *npr-1* mutants kept at 7% $O_2$ had much lower $Ca^{2+}$ in RMG compared to N2 animals kept at 21% $O_2$ (*Figure 2D*), they responded to RMG ChR2 activation significantly more strongly (*Figure 2E*). These results suggest that the inhibitory effects of *npr-1 215V* occur predominantly downstream of the $Ca^{2+}$ responses in RMG, possibly at the presynaptic level.

A network of G protein pathways modulates neurotransmission presynaptically: $G_q$ and $G_s$ signaling stimulates neurotransmitter release whereas $G_{o/i}$ signaling inhibits it, possibly by negatively regulating $G_s$ and $G_q$ signaling (*Miller et al., 1999*; *Nurrish et al., 1999*). In vitro work suggests that the NPR-1 215V and NPR-1 215F receptors can each couple to $G_{o/i}$ signaling following stimulation by FLP-21 (*Rogers et al., 2003*). If NPR-1 215V inhibits neurotransmission by activating $G_o$, then mutations that promote $G_q/G_s$ signaling over $G_o$ signaling should restore $O_2$ control of locomotory state and mimic *npr-1* mutants. The RGS (regulator of G protein signaling) protein EAT-16 terminates $G_q$ signaling by activating intrinsic $G_q$ GTPase activity (*Hajdu-Cronin et al., 1999*); RSBP-1 (R7 binding protein 1 homolog) interacts with and is required for EAT-16 activity (*Porter and Koelle, 2010*). Disrupting either *eat-16* or *rsbp-1* enabled N2 animals, which express *npr-1 215V*, to behave like *npr-1* mutants in response to a change in $O_2$ (*Figure 2F–H*). Together, these results suggest that a change in the balance of $G_o/G_q$ signaling in a subset of neurons explains how the NPR-1 215V neuropeptide receptor controls behavior.

## ASK, ASH, and ADL sensory neurons which have gap junctions with RMG are not required for the $O_2$-evoked switch in behavioral state

Animals integrate information across multiple sensory modalities to respond appropriately to changing environments. The ASK sensory neurons respond to pheromones and food (*Macosko et al., 2009*; *Wakabayashi et al., 2009*), and, like the URX $O_2$ sensors, make gap junctions with RMG (*White et al., 1986*; wormwiring.org; *Figure 1—figure supplement 1*). Previous work suggested that cGMP signaling in ASK is required for *npr-1* animals to move rapidly and to aggregate (*Tremain, 2004*; *Macosko et al., 2009*), leading to the hypothesis that ASK neurons are a major output of the RMG hub-and-spoke circuit (*Macosko et al., 2009*). The ASH and ADL nociceptive neurons also make gap junctions with RMG (*Figure 1—figure supplement 1*). Previous work suggested that TRPV signaling in ASH and ADL promotes aggregation behavior and avoidance of high $O_2$ (*de Bono et al., 2002*; *Chang et al., 2006*; *Rogers et al., 2006*). Together, these results suggested that gap-junctional communication across the RMG circuit integrates multiple sensory cues and is necessary for *npr-1* animals to move rapidly at 21% $O_2$ and to aggregate (*Macosko et al., 2009*).

To investigate this model in the context of $O_2$-evoked responses, we used the YC3.60 reporter to examine if changing $[O_2]$ altered $[Ca^{2+}]$ in ASK or ASH neurons in *npr-1* animals. A switch from 7% to 21% $O_2$ elicited a small increase in YFP/CFP FRET in ASK and ASH, indicating a rise in $[Ca^{2+}]$ (*Figure 3A,B*). The responses were sustained while animals were at 21% $O_2$. Deleting *gcy-35*

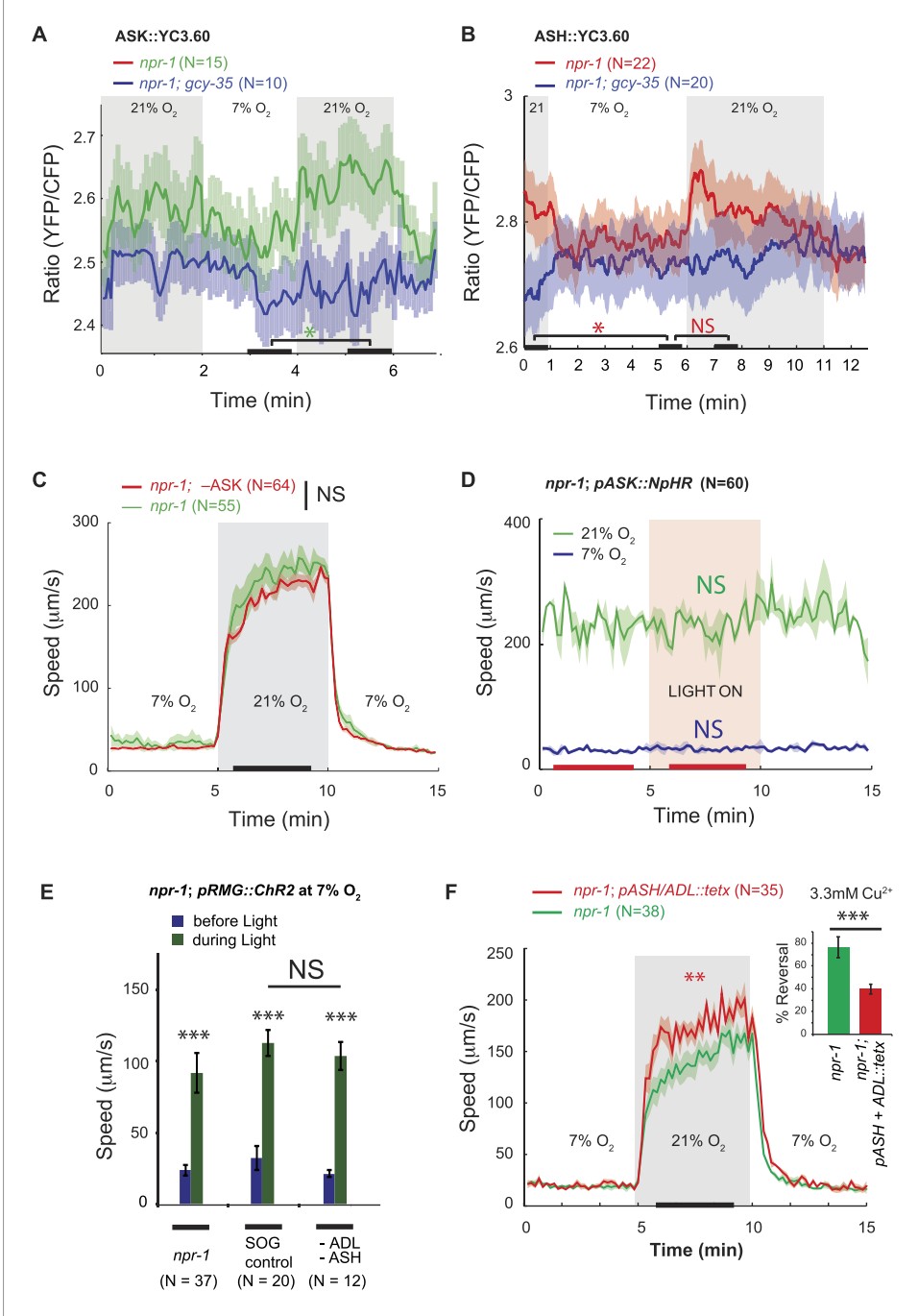

**Figure 3**. ASK, ASH, and ADL neurons are not necessary for the RMG circuit to stimulate rapid movement at 21% $O_2$. (**A** and **B**) In *npr-1* animals a 7% $O_2$ to 21% $O_2$ stimulus evokes a small rise in $Ca^{2+}$ in ASK (**A**) and ASH (**B**) neurons. These responses are abolished in *gcy-35*; *npr-1* animals. (**C** and **D**) Ablating ASK (**C**), or acutely inhibiting its activity using halorhodopsin (**D**) did not alter the locomotory behavior of *npr-1* animals on food at 7% or 21% $O_2$. (**E**) Stimulating RMG using ChR2 can stimulate locomotion in *npr-1* animals kept at 7% $O_2$ in the absence of ASH and ADL neurons. (**F**) Inhibiting synaptic release from ASH and ADL using tetanus toxin disrupts avoidance of $Cu^{2+}$ (inset) but does not inhibit the behavioral state switch evoked by changing $O_2$.

The following figure supplement is available for figure 3:

**Figure supplement 1**. Ablating ASK neurons does not reduce aggregation behavior of *npr-1* animals.

abolished the $O_2$-evoked $Ca^{2+}$ responses in ASK and ASH (*Figure 3A,B*), consistent with these responses being driven by URX, and therefore potentially via RMG. These results indicate that $O_2$ input can indeed modify ASK and ASH $Ca^{2+}$ levels, but the effect is small—at least at the cell body, where we made our measurements.

To ask if ASK neurons contributed to $O_2$-evoked behavioral states, we ablated ASK in *npr-1* animals by targeted expression of the *egl-1* cell death gene (*Wakabayashi et al., 2004*). ASK ablation was confirmed by dye filling and using a *psra-9::YC3.60* fiduciary marker. *npr-1* animals lacking ASK neurons responded to a 7–21–7% $O_2$ regime indistinguishably from control *npr-1* animals (*Figure 3C*). Optogenetic inhibition of ASK, in an *npr-1* strain expressing halorhodopsin (NpHR) specifically in ASK, had no effect on speed (*Figure 3D*). Ablating ASK neurons in *npr-1* animals also did not impair aggregation behavior (*Figure 3—figure supplement 1*). Aggregation is highly sensitive to $O_2$ circuit function and is disrupted in *gcy-35* mutants (*Cheung et al., 2004*; *Gray et al., 2004*; *Rogers et al., 2006*). These data suggest that ASK neurons are not necessary for $O_2$-evoked behavioral responses, or for aggregation.

To investigate if ASH and ADL are required for the $O_2$-controlled behavioral state switch, we ablated these neurons in *npr-1* animals expressing channelrhodopsin in RMG, using targeted expression of a miniSOG gene and light-induced ablation (*Qi et al., 2012*). The ASH and ADL ablated animals responded to high and low $O_2$ and to current injection into RMG like *npr-1* controls (*Figure 3E*). Tetanus toxin in the ASH and ADL neurons also did not disrupt $O_2$ responses (*Figure 3F*). To monitor toxin expression, we used a polycistronic construct that also expressed RFP, and we confirmed that tetanus toxin disrupted synaptic release from ASH and ADL by monitoring reversals in response to a 3 mM $Cu^{2+}$ drop (*Figure 3F*, inset). Thus, ablating ASK alone, or ASH and ADL together, does not disrupt relay of $O_2$-modulated RMG activity to downstream circuits that promote rapid movement.

## OCR-2 TRPV signaling is necessary in ASH, ADL, or ADF neurons to promote the $O_2$-evoked behavioral state switch

The ASH and ADL neurons express the TRPV1 homolog *ocr-2*, and disrupting *ocr-2*, or its partner subunit *osm-9*, attenuates animals' ability to navigate spatial $O_2$ gradients and to aggregate (*de Bono et al., 2002*; *Chang et al., 2006*; *Rogers et al., 2006*). *ocr-2*; *npr-1* animals only weakly modulated their locomotory activity when switched between 7% and 21% $O_2$ (*Figure 4A*). Expressing *ocr-2* cDNA in ASH or ADL or the serotonergic ADF neurons increased the speed of *ocr-2*; *npr-1* animals at 21% $O_2$ and, as expected, rescued the bordering and aggregation phenotype (*Figure 4A*, *Figure 4—figure supplement 1A*) (*de Bono et al., 2002*; *Chang et al., 2006*). The ability of *ocr-2* expression in ASH or ADL neurons to rescue the speed response and aggregation behavior of *ocr-2*; *npr-1* animals was retained when neurotransmission from these neurons was blocked using tetanus toxin (*Figure 4B,C*, *Figure 4—figure supplement 1B*). These data suggest that *ocr-2* TRPV activity in ASH and ADL is unlikely to promote $O_2$ responses by facilitating these neurons' neurosecretory activity, but may instead influence their gap junctional communication, for example, with RMG.

## Artificially stimulating ASH can rescue *ocr-2* TRPV channel defects

The OCR-2/OSM-9 TRPV channel is required for the ASH and ADL neurons to respond to sensory inputs (*Hilliard et al., 2004*). If disrupting *ocr-2* reduces ASH and ADL tonic or chronic activity, it could decrease RMG activity by reducing current input into RMG, or by shunting current away from RMG through the ASH–RMG and ADL–RMG gap junctions (*Figure 1—figure supplement 1*). If this model is correct, injecting current into ASH or ADL using ChR2 should restore to *ocr-2*; *npr-1* mutants rapid movement at 21% $O_2$.

To test this, we expressed ChR2 specifically in ASH by using the FLP/FRT system and by monitoring expression using YFP-tagged ChR2 (see 'Materials and methods'). We exposed transgenic animals to constant blue light of different intensities, ranging from 0.005 to 0.05 mW/mm$^2$. As expected, control animals not exposed to blue light did not strongly modulate their speed when switched between 7% and 21% $O_2$ (*Figure 4D*). By contrast, animals exposed to 0.005, 0.015 or 0.05 mW/mm$^2$ of continuous blue light modulated their locomotory state according to $O_2$ levels (*Figure 4D*). The amplitude of the $O_2$-evoked change in locomotory activity was similar to that obtained when we selectively restored *ocr-2* expression to ASH in *ocr-2*; *npr-1* animals (compare *Figure 4C,D*). Thus, injecting a constant $Ca^{2+}/Na^+$ current into ASH neurons is sufficient to restore $O_2$ modulation to the circuit in *ocr-2*; *npr-1* animals. This suggests that the effects of the *ocr-2* mutation are not developmental, as

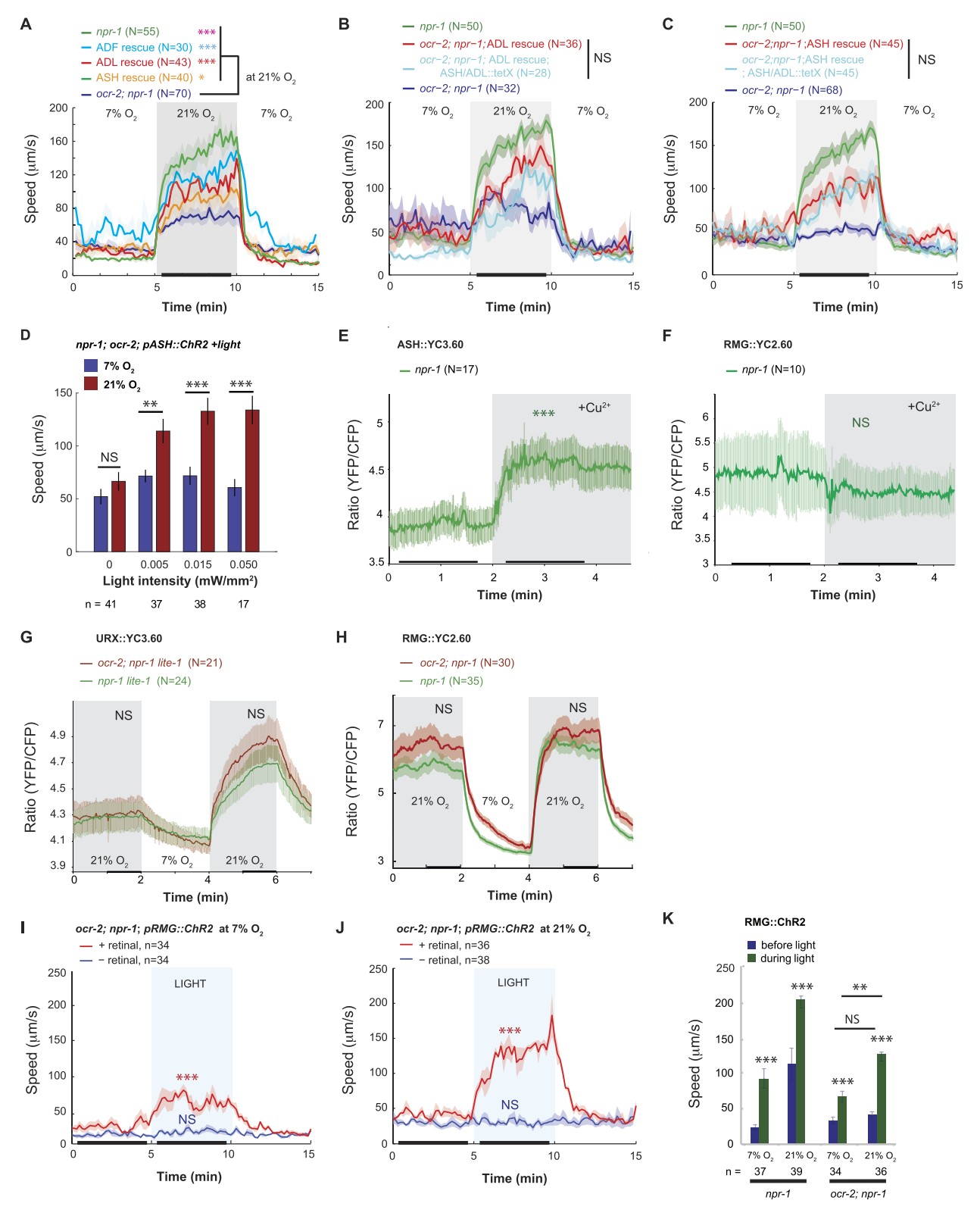

**Figure 4**. Disrupting a TRPV channel in ASH, ADL, and ADF neurons attenuates locomotory responses to 21% $O_2$. (**A**) The switch to rapid movement evoked in *npr-1* animals by 21% $O_2$ is attenuated in the absence of the OCR-2 TRPV channel. This defect of *ocr-2; npr-1* animals can be rescued to varying extents by expressing *ocr-2* cell-specifically in ASH, ADL, or ADF neurons. (**B** and **C**) The ability of *pADL::ocr-2* (**B**) or *pASH::ocr-2* (**C**) transgenes to rescue

*Figure 4. Continued*

the *ocr-2; npr-1* O$_2$ phenotype does not depend on synaptic release. Synaptic release was inhibited by expressing tetanus toxin in ASH and ADL neurons, using the *gpa-11* promoter. (**D**) Channelrhodopsin stimulation of ASH neurons restores modulation of locomotory activity by 21% O$_2$ to *ocr-2; npr-1* animals. (**E** and **F**) 2 mM Cu$^{2+}$ elicited strong Ca$^{2+}$ responses in ASH neurons of N2 and *npr-1* animals (**E**) but did not alter Ca$^{2+}$ in the RMG neurons (**F**). (**G** and **H**) Disrupting *ocr-2* did not significantly reduce O$_2$-evoked Ca$^{2+}$ responses in the cell bodies of URX (**G**) or RMG (**H**) neurons. (**I–K**) Stimulating RMG in *ocr-2; npr-1* animals kept on food using channelrhodopsin can partly restore O$_2$ modulation of locomotory activity.

The following figure supplement is available for figure 4:

**Figure supplement 1**. OCR-2 expression in ASH or ADL can restore aggregation behavior to *ocr-2; npr-1* animals (**A**) even when synaptic transmission in these neurons is inhibited by tetanus toxin expression (**B**).

they can be rescued in adult animals. Our results support a model in which tonic or chronic ASH (and ADL) activity facilitates O$_2$-evoked behavioral switches either by injecting current into RMG or by reducing shunting from the RMG circuit.

## TRPV channel activity alters O$_2$ circuit function upstream of NPR-1

A rise in O$_2$ evoked a small increase in ASH Ca$^{2+}$ (*Figure 3B*), suggesting that depolarizing current flows from RMG to ASH in *npr-1* animals at 21% O$_2$. To test explicitly if current can flow in the reverse direction, from ASH to RMG, we asked if strongly activating ASH using a noxious stimulus evoked a rise in RMG Ca$^{2+}$. As expected, a 10 mM Cu$^{2+}$ stimulus resulted in a large, sharp rise in ASH Ca$^{2+}$ that was easily detectable using YC3.60 (*Hilliard et al., 2004*) (*Figure 4E*). By contrast, this stimulus failed to evoke any Ca$^{2+}$ responses in the RMG cell body that could be measured with YC2.60 (*Figure 4F*). These results suggest that the anatomically defined gap junctions may not allow significant Ca$^{2+}$ current to flow from the ASH to the RMG cell body under our imaging conditions. How, then, does OCR-2 promote RMG activity? An alternative model is that by tonically/chronically depolarizing ASH and ADL neurons, OCR-2 reduces the current flowing from RMG to ASH and ADL via gap junctions. In *ocr-2; npr-1* mutants, a more negative membrane potential in ASH and ADL leads to more current being shunted from RMG to the nociceptive neurons, reducing RMG activity. To test this hypothesis, we compared O$_2$-evoked Ca$^{2+}$ responses in URX and RMG in *npr-1* and *ocr-2; npr-1* animals (*Figure 4G,H*). Surprisingly, the steady state Ca$^{2+}$ levels in the URX or RMG cell bodies were not significantly affected by the *ocr-2* mutation. Since the tetanus toxin experiments suggested that rescue of the *ocr-2; npr-1* phenotype by a *pASH::ocr-2* transgene did not require synaptic transmission, we speculate that the Ca$^{2+}$ effects of disrupting *ocr-2* are local to the gap junctions and not visible at the RMG cell body. Alternatively, ASH and ADL have the potential to leach away some other excitatory factor from RMG via gap junctions, and OCR-2 activity in ASH and ADL can attenuate this.

If loss of the OCR-2 TRPV channel somehow reduced the functionality of RMG, injecting current directly into RMG should rescue the phenotype of *ocr-2; npr-1* animals. Consistent with this, ChR2-activation of RMG in *ocr-2; npr-1* animals stimulated rapid movement, and inhibited reversals, both at 21% and 7% O$_2$ (*Figure 4I,J* and data not shown). This contrasts with the failure of ChR2-driven RMG activation to drive tonic changes in behavioral state in N2 animals (which express the NPR-1 215V receptor) (*Figure 2F*). These results suggest that the OCR-2-expressing neurons can facilitate RMG signaling in a way that is upstream of NPR-1 signaling.

## Neuropeptide transcription is coupled to peptide release and provides a readout of circuit state

Genetically encoded Ca$^{2+}$ sensors are blind to signaling mechanisms that modulate neurosecretion without altering Ca$^{2+}$ (*Miller et al., 1999*; *Nurrish et al., 1999*; *Rhee et al., 2002*; *Oda et al., 2011*). This limitation prompted us to seek a way to monitor enduring changes in neurosecretion as readouts of different global states. In vertebrates, several studies have reported that increased neural activity is associated with increased neuropeptide gene transcription (*Uhl and Nishimori, 1990*). In pancreatic β cells, feedback mechanisms couple insulin production and release (*Borgonovo et al., 2006*). If transcription of most neuropeptide genes is coupled to peptide release, *promoter::GFP* reporters for such genes should provide a readout of peptidergic circuit activity. To test this, we first studied

a *pflp-11::GFP* reporter expressed in the URX, AUA, and SAB neurons (*Kim and Li, 2004*). URX and AUA both show tonically elevated $Ca^{2+}$ at 21% $O_2$ compared to 7% $O_2$ (*Busch et al., 2012*). Cultivating animals in 7% $O_2$, or deleting *gcy-35*, reduced *pflp-11::gfp* expression in both URX and AUA (*Figure 5A*). By contrast, knocking out *ocr-2* did not, suggesting that TRPV signaling did not alter URX or AUA peptide expression (*Figure 5A*). The genotype at the *npr-1* locus did not affect *pflp-11::gfp* expression in URX, but expression in the AUA neurons was reduced in animals encoding the *npr-1 215V* allele compared to *npr-1* null mutants (*Figure 5A*). *pflp-11::gfp* expression in SAB, a neuron not known to be modulated by $O_2$, was not affected either by $O_2$ experience or the genotypes tested, providing an internal control. These data are consistent with *pflp-11* expression reporting neural activity.

To test more explicitly if modulation of *pflp-11::gfp* expression levels was related to neuron depolarization state, we inhibited URX by expressing a constitutively active $K^+$ channel related to *Drosophila ether-a-go*, EGL-2(GF) (*Weinshenker et al., 1999*). This transgene abolishes behavioral responses evoked by 21% $O_2$ (*Cheung et al., 2005*). EGL-2(GF) expression strongly reduced *pflp-11::gfp* expression in URX (*Figure 5B*). We next tested if feedback control of *pflp-11::gfp* expression was coupled to peptide release downstream of $Ca^{2+}$. To selectively block neurosecretion from URX, we cell-specifically expressed tetanus toxin, which cleaves synaptobrevin (*Schiavo et al., 2000*). Tetanus toxin expression strongly reduced *pflp-11::gfp* expression in URX, and the effect was stronger when toxin expression was higher (*Figure 5C*). These results suggest that *pflp-11::gfp* expression is coupled to peptide release in URX, and that peptide gene reporters may be useful surrogates to monitor long-term neurosecretory activity.

## $O_2$ levels and NPR-1 and OCR-2 signaling evoke widespread and long-lasting changes in circuit state

To test this hypothesis further, we examined a *pflp-5::GFP* transgene expressed in the RMG, ASE, and M4 neurons (*Kim and Li, 2004*). *pflp-5::gfp* expression in RMG was higher in *npr-1* animals grown at 21% $O_2$ than in *npr-1* animals grown at 7% $O_2$ or *gcy-35*; *npr-1* mutants grown at 21% $O_2$ (*Figure 5D*). These data suggest that *pflp-5::gfp* reporter expression is also coupled to neural activity. In contrast, expression of *pflp-5::GFP* in the M4 pharyngeal neuron, which is not known to respond to $O_2$, was not altered by $O_2$ experience or by disrupting *gcy-35*. To explore if *pflp-5* expression in RMG, like *pflp-11* expression in URX, was coupled to peptide release, we blocked RMG neurosecretion by cell-specifically expressing tetanus toxin. Animals expressing tetanus toxin in RMG showed significantly less *pflp-5* expression in RMG than non-expressing siblings (*Figure 5E*), consistent with neurosecretion feeding back to stimulate *pflp-5* transcription.

Expression of *pflp-5::gfp* in RMG was reduced in N2 animals compared to *npr-1* mutants, both at 21% and 7% $O_2$. Since at 21% $O_2$ RMG $Ca^{2+}$ levels were not strikingly different between N2 and *npr-1* mutants (*Figure 2E*), our data support a model in which NPR-1 215V signaling acts either downstream of $Ca^{2+}$ or locally to inhibit RMG neurosecretion. Interestingly, *ocr-2*; *npr-1* animals kept at 21% $O_2$ also showed reduced *pflp-5::gfp* expression compared to *npr-1* worms kept at the same $O_2$ levels (*Figure 5D*), suggesting that disrupting *ocr-2* reduced RMG neurosecretion in *npr-1* animals.

We next studied expression from a long version of the *flp-21* promoter that drives GFP expression in URA, M4, M2, ASJ, RMH, and RMG (*Macosko et al., 2009*). The RMH interneurons make gap junctions with RMG (*White et al., 1986*; wormwiring.org). Expression of the long *pflp-21::GFP* transgene in RMG and RMH was highest in *npr-1* animals cultivated at 21% $O_2$, and reduced in *gcy-35*; *npr-1* and *ocr-2*; *npr-1* animals grown at 21% $O_2$, as well as *npr-1* animals kept overnight at 7% $O_2$ (*Figure 5F*). *pflp-21::GFP* transgene expression in RMG and RMH was reduced in N2 animals compared to *npr-1* mutants, both at 21% and 7% $O_2$, supporting a model in which NPR-1 215V signaling inhibits neurosecretion from these neurons (*Figure 5F*). Thus, expression in RMG neurons driven from the long *flp-21* promoter and the *flp-5* promoter is regulated very similarly by $O_2$ experience and genotype at the *npr-1* and *ocr-2* loci. Together, our data suggest that neurosecretion from RMG and RMH is tonically modulated by $O_2$ levels and TRPV signaling.

A shorter version of the *flp-21* neuropeptide gene promoter drives GFP expression in the ADL, ASH, ASJ, URA, M4, and M2 neurons (*Kim and Li, 2004*). We wondered if this promoter could provide readouts of ASH and ADL signaling. Disrupting the *ocr-2* TRP channel subunit strongly reduced expression of the short *pflp-21::gfp* transgene in ASH and ADL in *npr-1* animals (*Figure 5G*). For ASH, we showed that expressing tetanus toxin also reduced *pflp-21::gfp* expression in this neuron

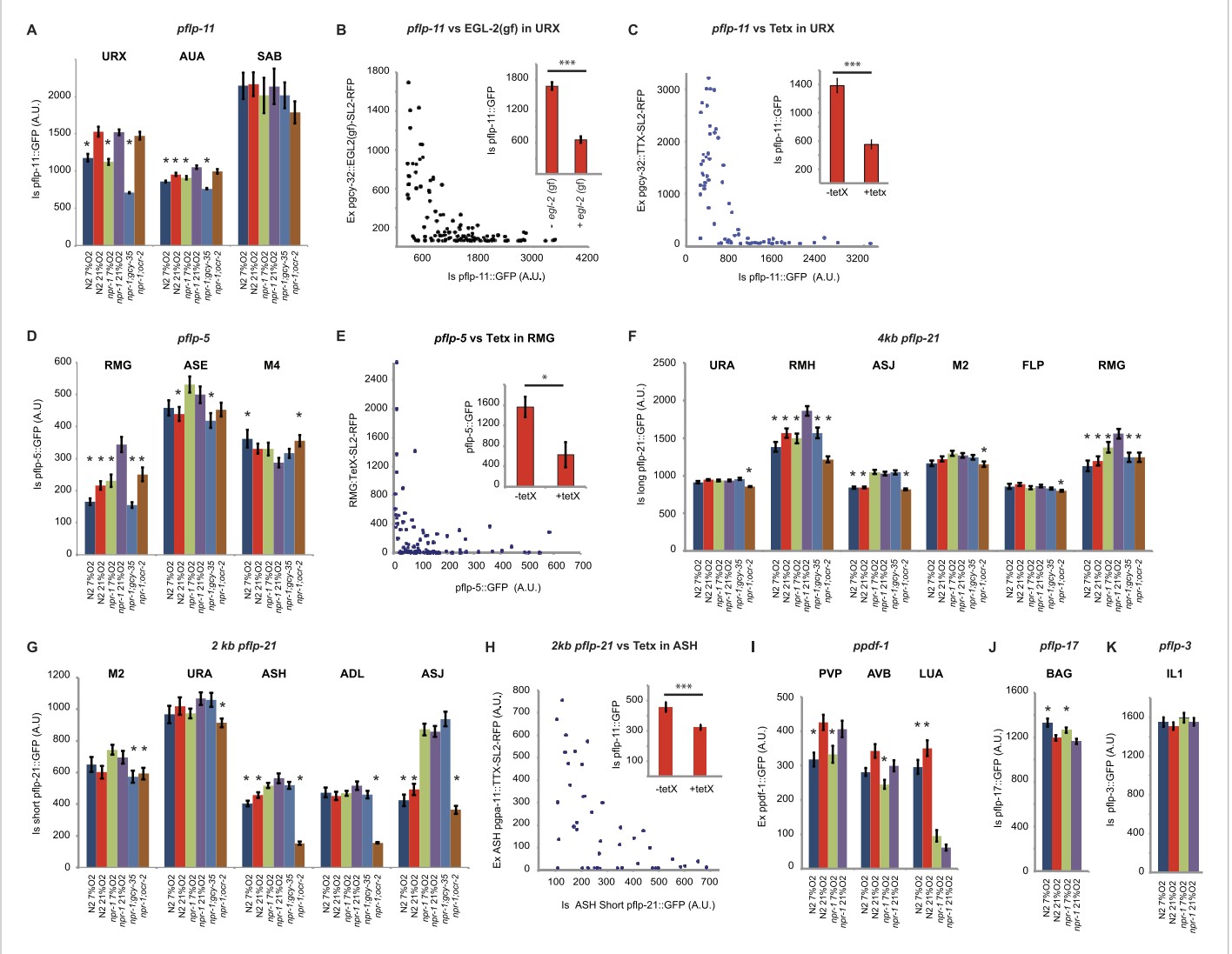

**Figure 5**. Neuropeptide gene expression levels report neurosecretory activity. (**A**) Expression of a *pflp-11* reporter in URX, AUA, and SAB neurons in different genotypes and at different $O_2$ tensions. In this and subsequent panels, AU = arbitrary units. (**B**) Expression of *pflp-11::GFP* in URX is inversely related to expression of a gain-of-function $K^+$ channel, EGL-2(GF), in the same neuron. Expression of the $K^+$ channel can be tracked due to co-expression of RFP in an operon. (**C**) Blocking exocytosis from URX by targeted expression of tetanus toxin strongly reduces expression of *pflp-11::GFP* in URX. Expression of tetanus toxin can be tracked due to co-expression of RFP in an operon. (**D**) Expression of a *pflp-5* reporter in RMG, ASE, and M4 in different genotypes and at different $O_2$ tensions. (**E**) Blocking exocytosis from RMG by targeted expression of a tetanus toxin strongly reduces expression of *pflp-5::GFP* in RMG. Expression of tetanus toxin can be tracked due to co-expression of RFP in an operon. (**F**) Expression of a *pflp-21* reporter that includes 4 kb of upstream sequences in URA (probably), RMH, ASJ, M2, FLP, and RMG neurons in different genotypes and at different $O_2$ tensions. (**G**) Expression of a *pflp-21* reporter that includes 2 kb of upstream sequences in M2, URA (probably), ASH, ADL, and ASJ neurons in different genotypes and at different $O_2$ tensions. (**H**) Blocking exocytosis from ASH by targeted expression of a tetanus toxin reduces expression of *pflp-21 (2 kb)::GFP* in ASH. Expression of tetanus toxin can be tracked due to co-expression of RFP in an operon. (**I–K**) Expression of *ppdf-1::gfp* (**I**), *pflp-17::gfp* (**J**) or *pflp-3::gfp* (**K**) reporters in different neurons in different genotypes at different $O_2$ concentrations. For (**A, D, F, G, I, J,** and **K**), asterisks indicate comparisons to *npr-1* animals kept at 21% $O_2$, *p<0.05; ***p<0.001.

(*Figure 5H*). These results are consistent with tonic/chronic OCR-2-dependent neurosecretory activity in these neurons driving a feedback loop to promote transcription of neuropeptides. Changing $O_2$ levels in *npr-1* animals did not significantly alter expression of this shorter *pflp-21::gfp* in ASH and ADL, suggesting that $O_2$ modulation of ASH and ADL secretion, if it occurs, is minor (*Figure 5G*). Note that *npr-1* is expressed in ASH neurons (*Figure 1—figure supplement 1*), which may account for differences in *pflp-21::gfp* expression in this neuron between *npr-1* and N2 animals. These data are

consistent with the ablation studies and the $Ca^{2+}$ imaging data for ASH, which suggest that ASH neurons exhibit only weak $O_2$-evoked changes in $Ca^{2+}$ and are dispensable for the $O_2$-evoked behavioral switch.

Encouraged by our results, we looked for other neuropeptide genes that are reported to be expressed in neurons connected to the $O_2$-sensing circuit by gap junctions or synapses. One such gene encodes the *C. elegans* ortholog of pigment dispersal factor, *pdf-1* (*Barrios et al., 2012*; *Meelkop et al., 2012*). A *ppdf-1::gfp* transgene is expressed in PVP and AVB, two neurons heavily connected with $O_2$-sensing neurons: PVP has gap junctions with AQR and PQR, whereas AVB has gap junctions with AQR and receives synaptic input from AQR, URX, and RMG. Expression of *ppdf-1::gfp* in both PVP and AVB was upregulated at 21% $O_2$ compared to 7% $O_2$, suggesting that the activity of these neurons is also modulated by $O_2$ (*Figure 5I* and see below). The expression of *ppdf-1::gfp* in RMG from the transgene was too weak and variable to be assayed.

Finally, we studied the expression of two additional peptide genes to demonstrate that not all neuropeptide promoters are stimulated by exposure to 21% $O_2$. The neuropeptide FLP-17 is expressed by BAG neurons and released when BAG is depolarized in vitro (*Ringstad and Horvitz, 2008*; *Smith et al., 2013*). BAG neurons are stimulated by low $O_2$ concentrations (*Zimmer et al., 2009*). As expected, *pflp-17::gfp* expression in BAG was stimulated at 7% $O_2$ compared to 21% $O_2$ (*Figure 5J*). The neuropeptide FLP-3 is expressed in the IL1 neurons (*Kim and Li, 2004*), which are part of a touch-responsive circuit, are poorly connected to known $O_2$-modulated neurons, and do not express NPR-1. *pflp-3::gfp* expression in IL1 was not regulated by $O_2$ or *npr-1* (*Figure 5K*).

Together, our data suggest that $O_2$ levels regulate *flp-11*, *flp-5*, *flp-21*, *flp-17*, and *pdf-1* peptide gene expression and peptide release in multiple neurons, including URX, AUA, RMG, RMH, BAG, PVP, and AVB, but not URA, SAB, IL1, M2, and M4. They also suggest that peptide release from RMG is stimulated by $O_2$-sensing neurons, depends on OCR-2 TRPV signaling, and is inhibited by NPR-1 215V signaling.

## Peptidergic signaling from RMG promotes rapid forward movement

Previous work has shown that RMG activity is necessary for high locomotory activity at 21% $O_2$ (*Macosko et al., 2009*; *Busch et al., 2012*). We have shown that RMG activity is sufficient to drive high locomotory activity, and that ASH, ADL, and ASK are each dispensable for these effects. Does peptidergic signaling from RMG contribute to driving high locomotory activity at high $O_2$? We first tested the effects of blocking all neurosecretion from RMG using cell-specific expression of a tetanus toxin transgene in *npr-1*. To monitor expression, we used a polycistronic construct that co-expressed the toxin with RFP. Animals expressing the transgene lost any tonic response to high $O_2$ and remained very poorly active at all $O_2$ concentrations (*Figure 6A,B*). This was expected from previous work (*Macosko et al., 2009*), although that study did not explicitly examine $O_2$-evoked responses. We next tested more specifically if neuropeptides secreted from RMG were required, using cell-specific RNAi to knock down the carboxypeptidase E (CPE) *egl-21* in RMG. Carboxypeptidase E removes C-terminal lysine and arginine residues from pro-peptides during maturation, and analysis of peptide extracts from *egl-21* mutants shows a deficit in the maturation of most *C. elegans* neuropeptides (*Husson et al., 2007*). *npr-1* animals with selective RNAi knockdown of *egl-21* in RMG moved appropriately slowly in 7% $O_2$, but sped up much less than non-transgenic siblings when switched to 21% $O_2$ (*Figure 6C*) and reversed more frequently (*Figure 6D*). These results indicate that neuropeptide release from the RMG neurons plays a major role in evoking the highly active state of feeding *C. elegans* at 21% $O_2$.

## Downstream circuitry

Ultimately, changes in $O_2$ levels modify behavioral state by altering motor circuits. We envisioned three ways by which $O_2$ circuit output might alter downstream circuits. In one model, the downstream circuits would exhibit $O_2$-evoked tonic changes in $Ca^{2+}$ levels, much as we observe in the URX, AQR, PQR, AUA, and RMG neurons (*Busch et al., 2012*). In a different model, behavioral state would be encoded across multiple neurons whose activity does not faithfully track $O_2$ concentration, but which on average show $O_2$-evoked changes in $Ca^{2+}$ levels. To investigate these possibilities, we used $Ca^{2+}$ indicators to image the activity of motoneurons, and of interneurons in layers upstream of motoneurons, at different $O_2$ environments. The third possibility would not involve regulation of $Ca^{2+}$ levels, but rather presynaptic effects that would only be observed using other reporters, such as the peptide promoter assay.

We focused our studies on four sets of neurons: the A and B motoneurons, which are thought to mediate reverse and forward movement, respectively; the AVA 'command' interneurons, which

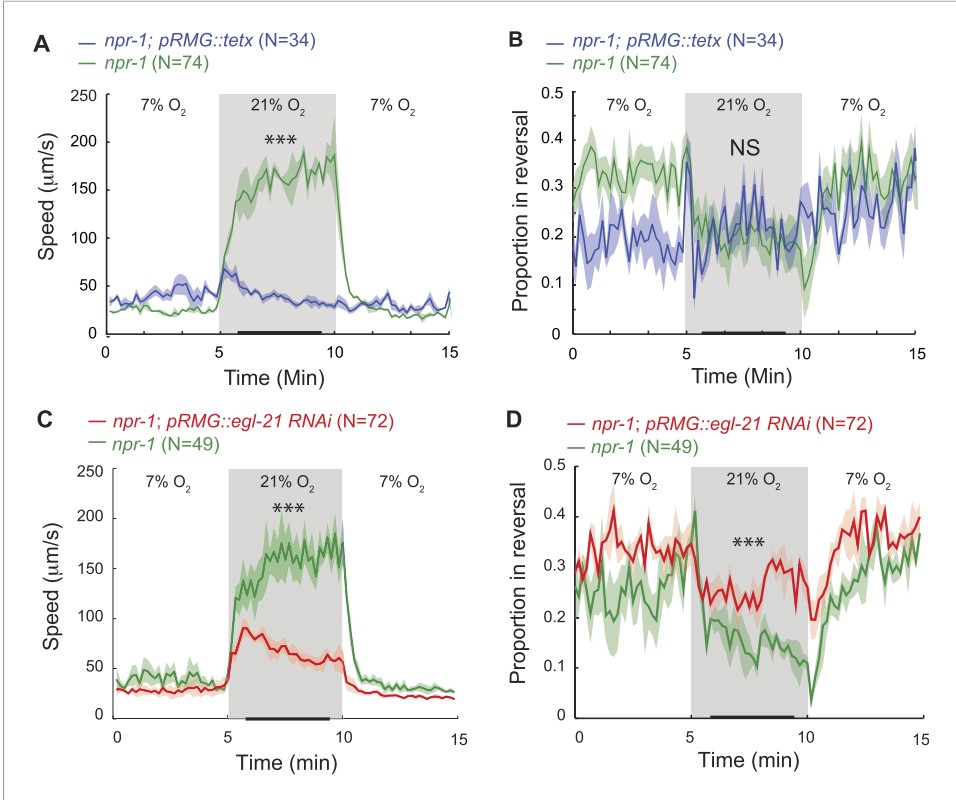

**Figure 6**. RMG neuropeptide secretion drives rapid movement at 21% $O_2$. (**A–D**) Selective expression of tetanus toxin (**A** and **B**) or RNAi knockdown of EGL-21 carboxypeptidase E (**C** and **D**) in RMG inhibits the $O_2$-evoked switch in locomotory state. NS, not significant; ***$p<0.001$.

promote backward movement by ensuring that the A motoneurons are more active than the B motoneurons; the AVB 'command' interneurons which promote forward movement by ensuring that the B motoneurons are more active than the A motoneurons; and the AIY interneurons which do not have anatomically defined connections with URX or RMG but are post-synaptic to multiple other sensory neurons (*White et al., 1976*, *1986*; *Chalfie et al., 1985*; *Kawano et al., 2011*; wormwiring. org; *Figure 1—figure supplement 1*). As a control, we imaged the RMG neurons.

We first measured spontaneous (*Figure 7A,B*) and $O_2$-evoked (*Figure 7C,D*) $Ca^{2+}$ changes in animals immobilized with 3 mM levamisole; levamisole immobilizes *C. elegans* by activating nicotinic acetylcholine receptors expressed in body wall muscle (*Lewis et al., 1980*). As expected (*Kawano et al., 2011*), $Ca^{2+}$ levels in AVA and AVB neurons, and in the A and B neurons, were anti-correlated. As observed previously (*Schrödel et al., 2013*), in unstimulated animals the bouts of activity in which neurons promoting backward movement (the AVA and VA neurons) were more active than neurons promoting forward movement (the AVB and VB neurons) often lasted for more than a minute before switching stochastically (*Figure 7A,B*). This fictive behavior contrasted with the real behavior of animals moving on an agar plate, which only reverse for a few seconds. Nevertheless, the fictive behavior evoked by a rise in $O_2$ recapitulated some of the expected response: the backward module (VA neurons) was reliably activated by $O_2$ upsteps from 7% to 21% $O_2$ (*Figure 7C*). Moreover, when we exposed animals to a graded $O_2$ series of 21–14–7–21%, we found that, despite frequent switches in activity, the average activity in AVB and AIY was modulated by $O_2$ levels, with higher $Ca^{2+}$ at 21% compared to 7% in both neurons (*Figure 7D*).

## Information coding by RMG, AVB, and AIY neurons

The AVA and AVB interneurons, and their downstream targets the A and B motoneurons, respond to a variety of sensory cues transmitted by upstream circuits to control *C. elegans*' direction of

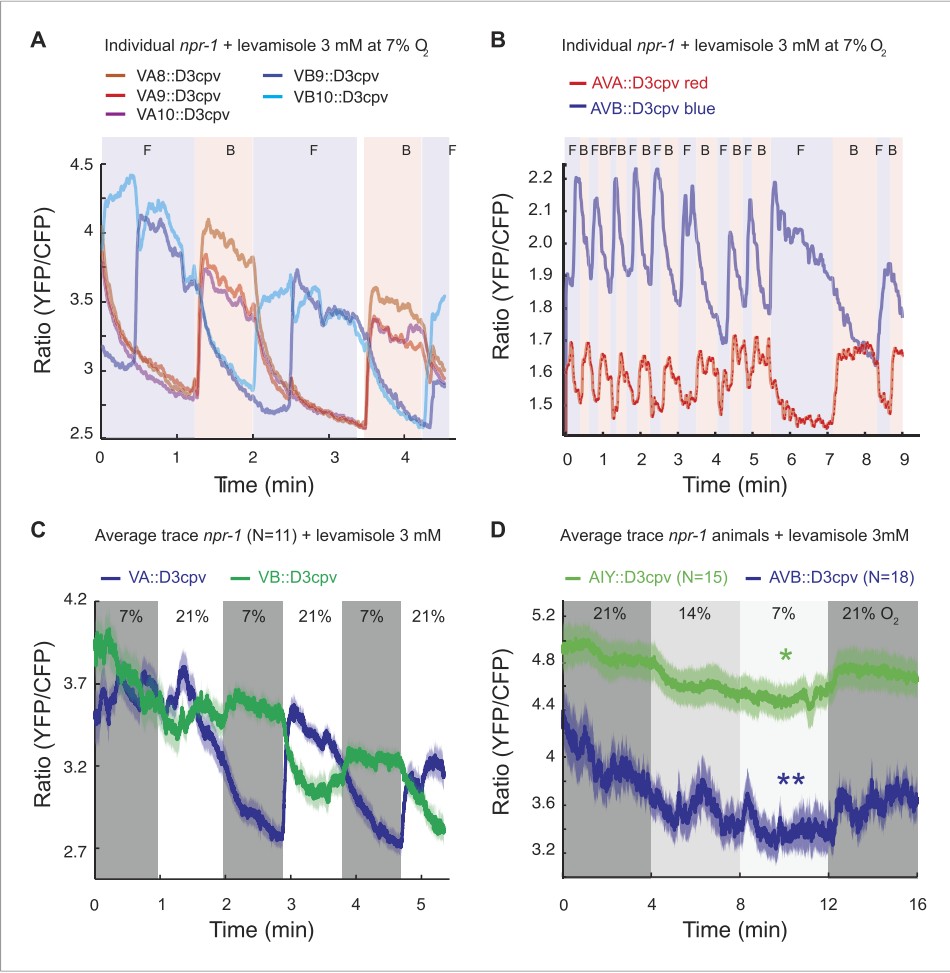

**Figure 7**. Spontaneous and $O_2$-evoked $Ca^{2+}$ responses in interneurons and motoneurons in immobilized animals. (**A** and **B**) Spontaneous bi-stable $Ca^{2+}$ responses observed in VA and VB neurons (**A**) and in AVA and AVB neurons (**B**) in individual *npr-1* animals immobilized with 3 mM levamisole and kept at 7% $O_2$. As expected, $Ca^{2+}$ responses in VAs versus VBs (**A**) and AVA versus AVB (**B**) neurons were anti-correlated. Different colored shading and F and B indicate fictive forward (F) or backward (B) movement. (**C**) On average, an upstep from 7% to 21% $O_2$ evoked a $Ca^{2+}$ response in the VAs neurons of *npr-1* animals immobilized with 3 mM levamisole. (**D**) Despite stochastic, high amplitude changes in $Ca^{2+}$ levels of AIY and AVB neurons, on average, higher $[O_2]$ correlated with higher $Ca^{2+}$ levels in AIY and AVB in animals immobilized with 3 mM levamisole.

movement. These upstream circuits include highly connected interneurons called AIA, AIB, and AIY that are post-synaptic to multiple sensory neurons (*White et al., 1986*; wormwiring.org). We speculated that the highly variable $Ca^{2+}$ baseline we observed in AVA, AVB, and AIY reflected these diverse inputs, and that modulation by $O_2$ levels acts on top of these other inputs. In levamisole-treated animals, the frequent switches observed in AVA, AVB, and AIY might reflect attempts by the animals to change their direction of movement.

To examine this possibility without the caveats associated with levamisole treatment, and in a way that allows us to record information about the animals' behavior, we imaged neural activity in animals moving freely on food under a layer of PDMS, at 7% or 21% $O_2$. Under these conditions, animals modulated their speed of movement in response to $O_2$ levels, although with reduced amplitude, but did not exhibit the high reversal state normally observed at 7% $O_2$, perhaps because of the effects of being under a PDMS layer (*Figure 8A*; *Figure 8—figure supplement 1*).

As expected from our imaging of immobilized animals, RMG interneurons responded to a 7–21% increase in $O_2$ with a large tonic rise in $Ca^{2+}$ that lasted for as long as animals were at 21% $O_2$

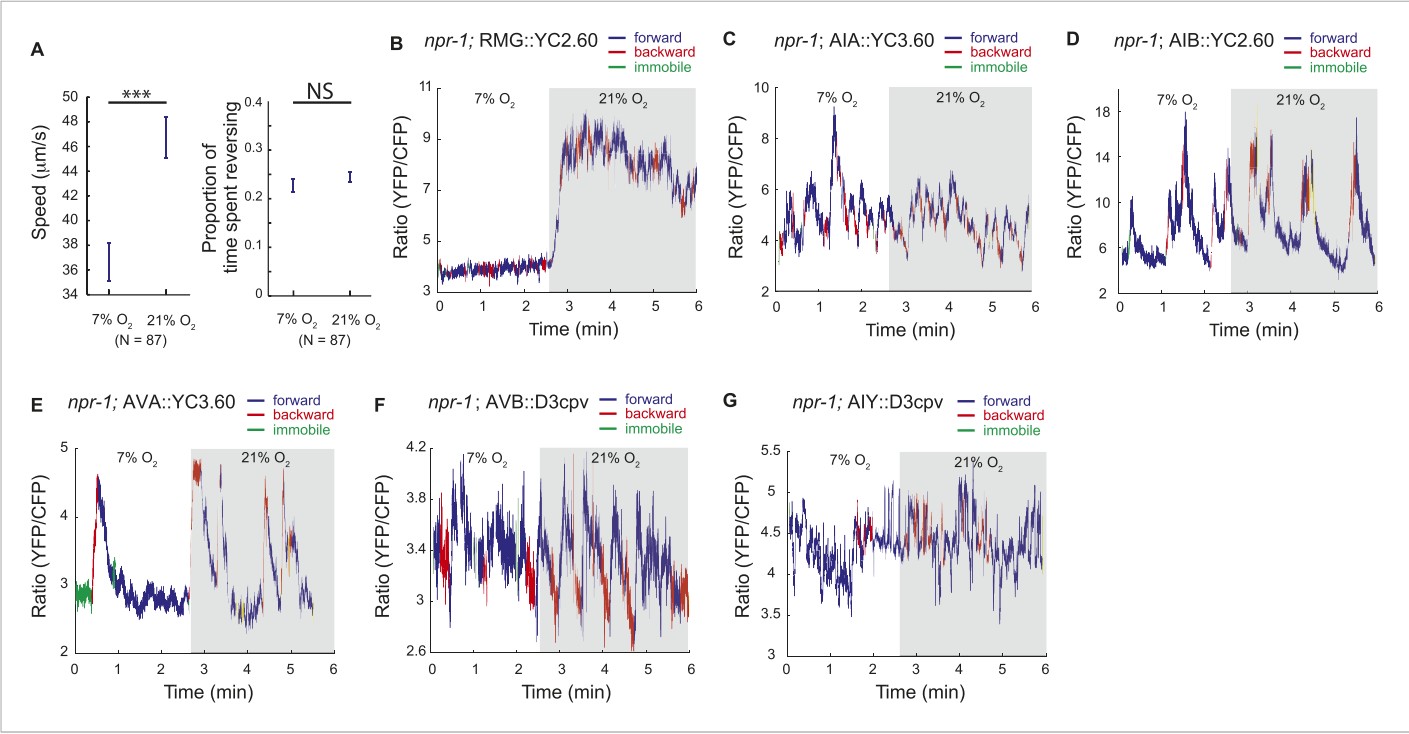

**Figure 8.** Correlation of neural activity and behavior at 7% and 21% $O_2$. (**A**) Under our $Ca^{2+}$ imaging conditions, freely moving *npr-1* animals increase their speed but do not suppress reversals at 21% $O_2$. (**B–G**) RMG neurons respond to 21% $O_2$ with a strong persistent increase in $Ca^{2+}$ regardless of direction of travel (**B**). By contrast, freely moving *npr-1* animals display frequent brief $Ca^{2+}$ changes in AIA (**C**), AIB (**D**), AVA (**E**), AVB (**F**), and AIY neurons (**G**), at both 7% and 21% $O_2$. Most $Ca^{2+}$ changes are associated with reversal events.

The following figure supplement is available for figure 8:

**Figure supplement 1.** The experimental setup used to image freely moving animals.

(*Figure 8B*). $Ca^{2+}$ levels in RMG did not change when animals reversed or stopped moving forward. By contrast, the AVB, AVA, AIY, AIA, and AIB interneurons each displayed frequent, large but brief changes in $Ca^{2+}$ levels, at both 21% and 7% $O_2$ (*Figure 8C–G*). Superficially, these $Ca^{2+}$ changes appeared to occur stochastically, as in immobilized animals, but careful analysis showed that they usually coincided with a switch in the direction of travel, and never lasted more than a few seconds (*Figure 8C–G*).

We hypothesized that the large changes in $Ca^{2+}$ levels associated with executing reversal behaviors would obscure any smaller $Ca^{2+}$ changes in these neurons that were evoked by a change in $O_2$ concentration. To examine this possibility, we registered $Ca^{2+}$ traces to the initiation of a reversal, and averaged responses across multiple animals kept at either 7% or 21% $O_2$. We then compared $Ca^{2+}$ levels in each neuron as animals executed reversals at 7% or 21% $O_2$ (*Figure 9A*). As expected, we observed changes in $Ca^{2+}$ associated with the reversal behavior sequence. Superimposed on this, our analysis revealed that $Ca^{2+}$ levels in AVB and AIY, but not in AIA, AIB, and AVA neurons, differed significantly between animals kept at 21% and 7% $O_2$ (*Figure 9B–F*). AVB showed higher $Ca^{2+}$ levels during forward locomotion at 21% $O_2$ than at 7% $O_2$ (*Figure 9B*). AIY also showed significantly higher $Ca^{2+}$ levels at 21% $O_2$ than 7% $O_2$ during forward locomotion (*Figure 9C*), and only showed changes in activity correlating with reversal behavior at 7% $O_2$. As expected, the RMG interneurons showed strong modulation by $O_2$ but no change in activity correlating with reversal state (*Figure 9G*). Together, our data predict that modulation of AVB and AIY downstream of RMG mediates behavioral changes induced by different $O_2$ environments.

Our data also suggest a sequence of changes in neural activity associated with the initiation and the termination of reversals at 7% and 21% $O_2$. In particular, we observed a fall in AIA $Ca^{2+}$ coincident with a rise in AIB $Ca^{2+}$ that immediately preceded reversal initiations, and was followed by a drop in AVB $Ca^{2+}$ and a rise in AVA $Ca^{2+}$ around the time reversals begin (*Figure 9H*). Termination of reversals was

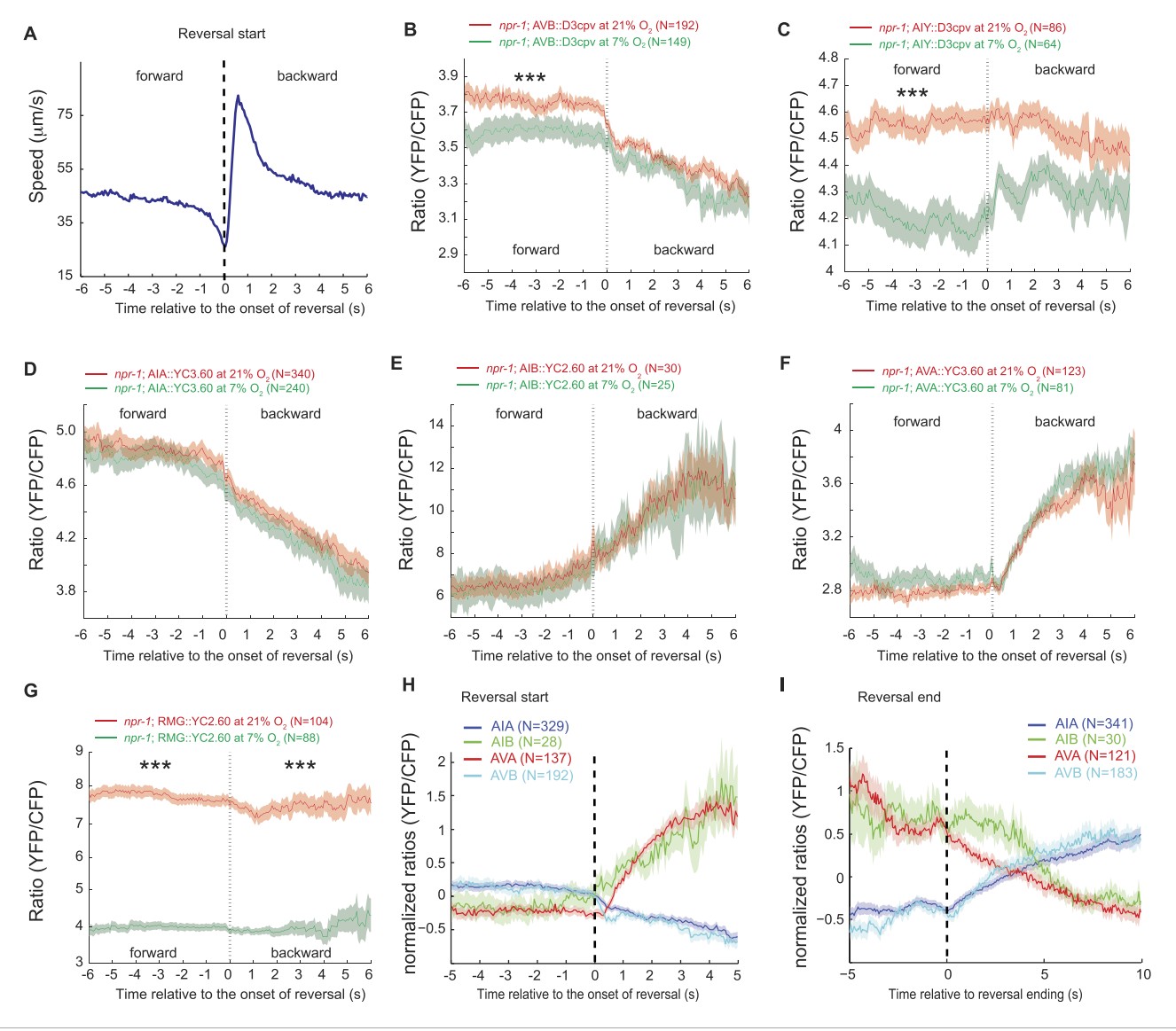

**Figure 9**. AVB and AIY interneurons integrate information about $O_2$ levels with other input. (**A**) Animals showed a characteristic pattern of speed changes when traces were aligned according to the time of reversal initiations. (**B–G**) AVB (**B**), AIY (**C**), and RMG neurons (**G**) show increased $Ca^{2+}$ at 21% $O_2$ compared to 7% $O_2$ during forward movement. By contrast, average $Ca^{2+}$ in AIA (**D**), AIB (**E**), and AVA (**F**) was not significantly modulated by $O_2$. (**H** and **I**) Normalized $Ca^{2+}$ traces in AVA, AVB, AIA, and AIB neurons aligned to the first frame of backward locomotion (**H**), or to the first frame of forward movement (**I**), during spontaneous reversals. Reversal initiation correlates with a rise in $Ca^{2+}$ in AIB and AVA and a fall in $Ca^{2+}$ in AIA and AVB. The converse pattern is observed when reversals are terminated.

preceded by a fall in AVA $Ca^{2+}$, and associated with a rise in $Ca^{2+}$ in AVB and AIA, and a drop in $Ca^{2+}$ in AIB (*Figure 9I*).

## Optogenetic control of AIA, AIY, and AVB can phenocopy $O_2$-evoked behavioral responses

To test for physiological roles of AIY, AVB, and AIA interneurons in $O_2$-evoked changes in behavioral state we turned to optogenetics and ablation experiments. We first expressed halorhodopsin (eNpHR2) in the AIY, AVB, and AIA interneurons, or in AIA alone, in an *npr-1* background, and used light to inhibit these neurons. At 7% $O_2$, acute inhibition of AIY, AVB and AIA together, or of AIA alone, only had minor effects on behavior (*Figure 10A*). When the light was turned off, we observed

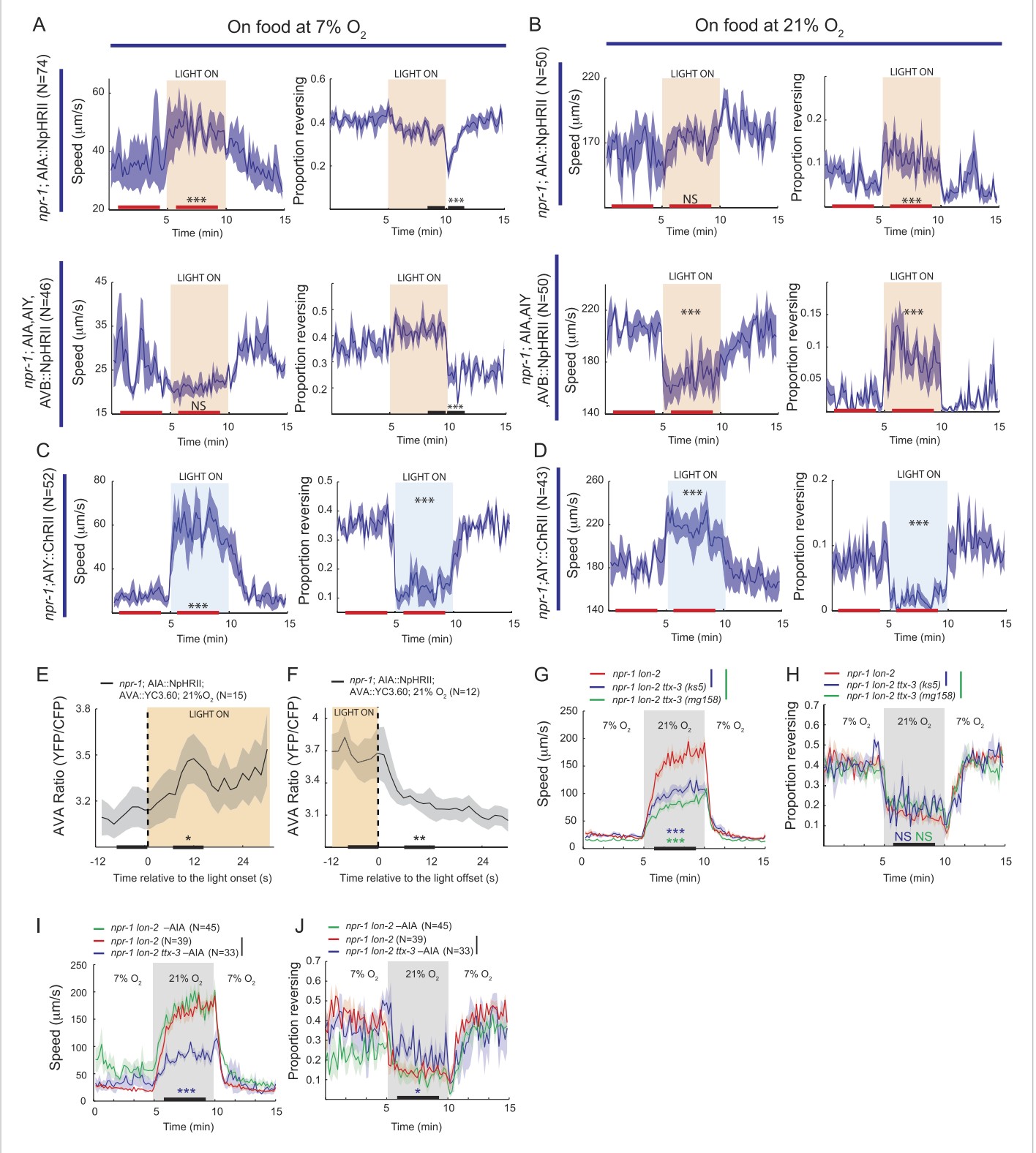

**Figure 10**. AIY and AVB interneurons contribute to the switch in locomotory activity evoked by 21% O$_2$. (**A** and **B**) Behavioral effects of inhibiting AIA or AIA, AIY and AVB using halorhodopsin in animals kept at 7% O$_2$ (**A**) or 21% O$_2$ (**B**). (**C** and **D**) Behavioral effects of activating AIY neurons using channelrhodopsin in animals kept at 7% O$_2$ (**C**) or 21% O$_2$ (**D**). (**E** and **F**) Ca$^{2+}$ responses evoked in AVA interneurons by inhibition (**E**) or disinhibition (**F**) of AIA neurons using halorhodopsin. (**G** and **H**) Disrupting the *ttx-3* homeobox transcription factor required to specify AIY cell fate attenuates the rapid
*Figure 10. continued on next page*

*Figure 10. Continued*

movement evoked in *npr-1* animals by 21% $O_2$ but not the inhibition of reversals. *ttx-3 (ks5)* and *ttx-3(mg158)* are different null alleles. (**I** and **J**) Ablating AIA neurons alone does not disrupt responses of *npr-1* animals to 21% $O_2$. *npr-1 ttx-3* mutants ablated for AIA neurons both move more slowly and reverse more frequently than controls.

a transient decrease in the rate of reversals that probably reflected disinhibition of AIA (*Figure 10A*, compare AIA alone with AIA, AIY, and AVB).

When animals were kept at 21% $O_2$, acute inhibition of AIA caused a sustained increase in reversal rate compared to control animals, but did not significantly alter their speed of movement (*Figure 10B*). Inhibition of AIA, AIY and AVB together at 21% $O_2$ caused a sustained decrease in the speed of movement together with an increased rate of reversals compared to control animals (*Figure 10B*, compare AIA alone with AIA, AIY, and AVB). Together, our data suggest that although we do not observe regulation of AIA $Ca^{2+}$ by $O_2$ (*Figure 9D*), tonic AIA activity plays a role in inhibiting reversal rate at 21% $O_2$, while AIY and AVB promote rapid forward movement.

Since inhibiting AIA at 21% $O_2$ promoted reversal behavior (and disinhibiting AIA suppressed reversals at 7% $O_2$), we examined how altering AIA activity modulated downstream circuits. We imaged $Ca^{2+}$ in the AVA interneurons of freely moving *npr-1* animals kept at 21% $O_2$ while acutely inhibiting or disinhibiting AIA with light. Inhibiting AIA evoked a rise in AVA $Ca^{2+}$ (*Figure 10E*), while disinhibiting AIA caused a rapid fall in AVA $Ca^{2+}$ (*Figure 10F*). These results suggest that AIA has inhibitory control of AVA activity, and is consistent with AIA activity preceding AVA activity during spontaneous reversals.

We next expressed ChR2 selectively in AIY in an *npr-1* background, and used light to activate this interneuron at low and high $O_2$ concentrations (*Figure 10C,D*). Activating AIY caused a sustained decrease in reversal rate and an increase in speed, both at 7% and 21% $O_2$. Thus, consistent with the predictions made from our $Ca^{2+}$ imaging experiments, optogenetics suggest that increasing AIY activity can increase the speed of locomotion and decrease reversal frequency.

To test further for a role of AIY in the $O_2$-evoked behavioral switch, we examined the $O_2$ responses of *npr-1 ttx-3* mutants. *ttx-3* encodes a LIM homeodomain transcription factor required for specification of AIY (*Hobert et al., 1997*). *ttx-3 npr-1* animals sped up less than *npr-1* animals upon being switched from 7% to 21% $O_2$, but inhibited reversal behavior to a similar extent (*Figure 10G,H*). Ablating AIA interneurons in *ttx-3 npr-1* animals did not alter speed further, but significantly increased reversal rate at 21% $O_2$ (*Figure 10I,J*). These results are consistent with AIY playing a physiological role in promoting rapid movement at 21% $O_2$, and suggest that AIA acts with AIY to inhibit reversal behavior at 21% $O_2$, although neither neuron appears to be essential.

## Discussion

We reverse engineer a neural circuit that controls the global state of *C. elegans*, enabling the animal to recognize 21% $O_2$ and avoid, escape, and adapt to surface exposure (*Figure 11*). the URX, AQR, and PQR $O_2$-sensing neurons are tonically stimulated by 21% $O_2$, causing sustained changes in *C. elegans* behavior, gene expression, and circuit state. The RMG interneurons, connected to URX by gap junctions and reciprocal synapses, are activated by URX and drive an aroused locomotory state that promotes escape from high $O_2$. Stimulating RMG using channelrhodopsin is sufficient to evoke rapid and sustained forward movement in animals kept at 7% $O_2$, even if URX, AQR, and PQR are ablated. However, stimulating RMG does not elicit the transient avoidance responses evoked when *C. elegans* encounter an increase from 7% to 21% $O_2$. Distinct circuits appear to coordinate the transient and persistent behavioral changes evoked by a rise in $O_2$.

RMG activation switches behavioral state in large part by tonic release of neuropeptides. Neuropeptides are used across a wide phylogenetic spectrum to create, maintain, or amplify neuronal network states (*Marder et al., 2014*). Multiple downstream neurons show long-lasting activity changes in response to the tonic information from URX, AQR, PQR, and RMG. The AIY and AVB interneurons act downstream of URX–RMG to promote rapid forward movement at 21% $O_2$. Unlike URX and RMG neurons, whose $Ca^{2+}$ levels tonically report $O_2$ levels, $Ca^{2+}$ levels in AIY and AVB vary widely and rapidly even when $O_2$ levels are unchanging. On average however, $Ca^{2+}$ in these neurons is higher at 21% $O_2$ than at 7% $O_2$. Thus, tonic input from $O_2$ circuits modulates but does not set AIY and AVB activity.

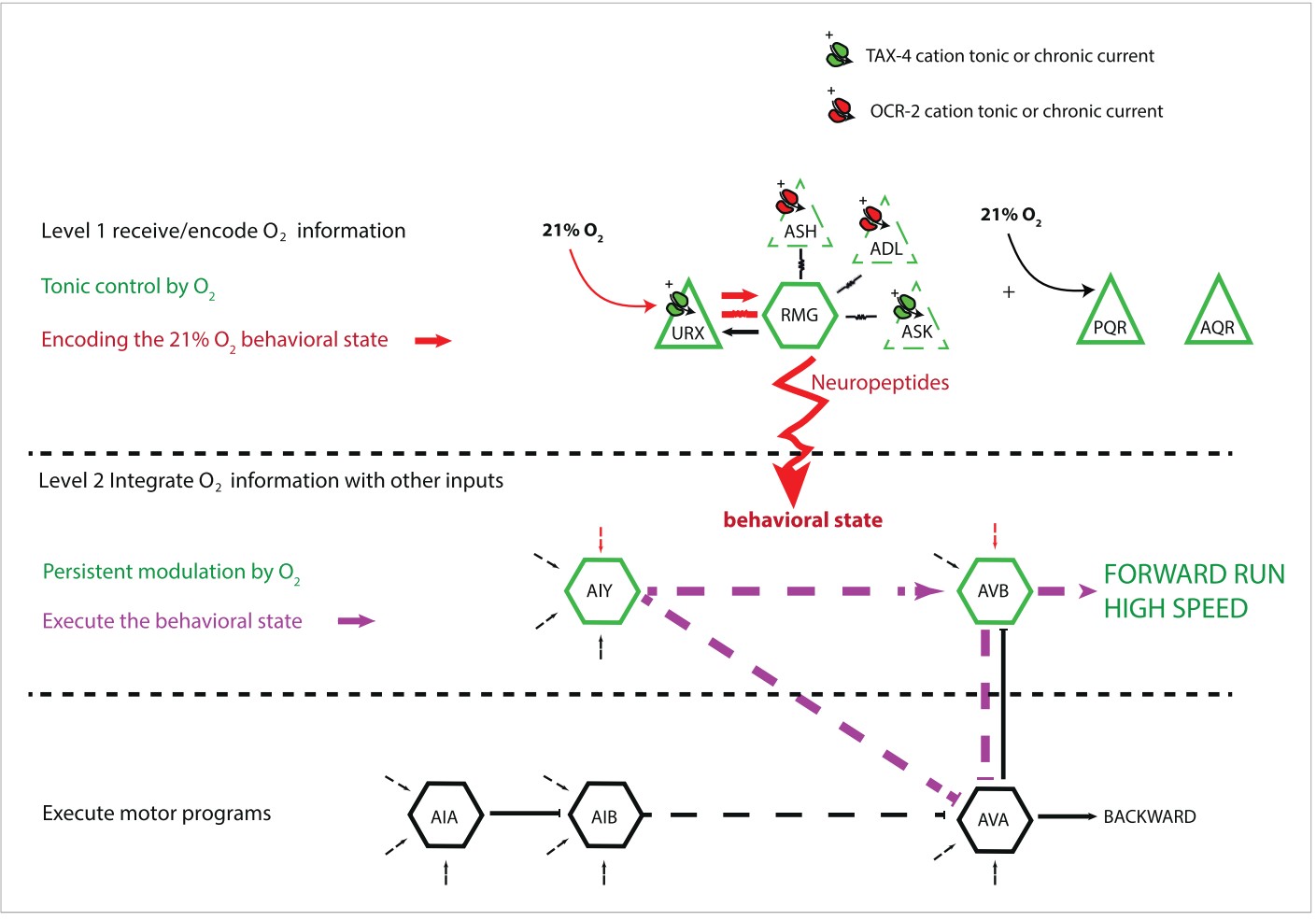

**Figure 11**. A model for the circuits controlling the behavioral switch. The URX $O_2$-sensing neurons and the RMG interneurons tonically encode $O_2$ concentration. These neurons communicate to downstream neurons predominantly through sustained changes in neuropeptide secretion. RMG is connected through gap junctions to sensory neurons that can act as shunts, downregulating RMG neurosecretion when the OCR-2 or TAX-4 cation channels they express are less active. URX–RMG output continuously modulates the activity of downstream neurons, including AIY and AVB, but these neurons also respond to other cues, and their activity at any time point reflects both $O_2$ concentration and the behavior being executed. Increased activity of these neurons promotes rapid forward movement. Other neurons, such as AIA and AVA, participate in the execution of the behavioral state but do not appear to be under tonic control of the URX–RMG circuit.

Rapidly varying $Ca^{2+}$ levels in AIY and AVB likely reflect the ability of these neurons to respond to inputs from other sensory circuits while receiving tonic signals from the $O_2$-sensing circuit (*Figure 11*).

## Neuropeptide gene transcription as readout of circuit activity

To delineate long-term changes in the activity of neural populations, we sought simple readouts of neurosecretory activity that do not depend on $Ca^{2+}$ sensors. $Ca^{2+}$ imaging approaches, although powerful, are blind to signaling mechanisms that alter synaptic and neurosecretory activity without affecting $Ca^{2+}$ levels. We were inspired by sporadic reports suggesting a positive correlation between neural activity and neuropeptide gene transcription in the mammalian brain. By studying multiple neuropeptide genes expressed in URX, RMG, and other neurons, we showed that transcription of these genes is positively coupled to the secretory activity of the neurons expressing them. The feedback loop appears to act downstream of $Ca^{2+}$, and to be linked to neurosecretion itself, since it can be disrupted by expressing tetanus toxin, which inhibits synaptic release by cleaving synaptobrevin. The effects of tetanus toxin on neuropeptide expression recapitulate the effects of manipulating circuit activity by disrupting signaling molecules or by altering $O_2$ levels. If our

observations are generalizable, neuropeptide transcriptional reporters provide a way to dissect tonic modulation of secretory activity in neural networks.

## The neuropeptide receptor NPR-1 inhibits RMG output downstream of $Ca^{2+}$

The ability of RMG interneurons to alter behavioral state according to ambient $O_2$ is inhibited by an NPY/RFamide-like neuropeptide receptor, NPR-1. The *npr-1* allele found in the N2 standard *C. elegans* laboratory strain, *npr-1 215V*, essentially abolishes RMG-mediated escape responses in feeding animals. Our data suggest *npr-1* acts by inhibiting neurosecretion from RMG downstream of $Ca^{2+}$. First, we do not observe a striking difference in $O_2$-evoked $Ca^{2+}$ responses in RMG between *npr-1* and *npr-1 215V* animals. Second, whereas ChR2 activation of RMG can induce *npr-1* animals kept at 7% $O_2$ to switch to rapid movement, it has little effect on animals encoding the NPR-1 215V receptor, even at 21% $O_2$ when RMG $Ca^{2+}$ levels are high. Third, transcriptional reporters of neuropeptide genes suggest that NPR-1 215V inhibits neurosecretion from RMG and other neurons. NPR-1 may inhibit synaptic release by altering the balance between $G_o$ and $G_q$ signaling in favor of $G_o$. Increasing $G_q$ activity, by disrupting the EAT-16 RGS protein or its binding protein RSBP-1, phenocopies the effects of disrupting *npr-1*. EAT-16 and RSBP-1 activate the $G_q$ GTPase.

## Hub-and-spoke model

RMG lies at the center of an anatomically defined hub-and-spoke network connected by gap junctions that include the ASK, ADL, ASH, and URX sensory neurons (*Figure 11*). Previous work suggested that the RMG hub redistributes inputs across the spoke sensory neurons through gap junctions to direct behavioral responses (*Macosko et al., 2009*). The importance of gap junctions in the circuit was inferred from the anatomy and genetic manipulations. Whether RMG could propagate electrical signals or $Ca^{2+}$ across the network, and if so in which direction, or if gap junctions were rectifying or passive gates was unknown. Our data revise our understanding of how this circuit works, as detailed below.

## ASK is not a necessary output for the $O_2$-evoked behavioral switch

ASK are 'OFF' neurons that act analogously to vertebrate photoreceptors, responding to ascaroside pheromones or food stimuli with a decrease in $Ca^{2+}$ (*Macosko et al., 2009*; *Wakabayashi et al., 2009*). ASK neurons were proposed to be a major output of the RMG hub-and-spoke network, inducing rapid movement at 21% $O_2$ and promoting aggregation in response to pheromones (*Macosko et al., 2009*). In this model, RMG stimulates ASK excitability via gap junctions in aggregating strains, but in solitary strains this is prevented because NPR-1 215V directly or indirectly inhibits gap junctional signaling in the network. We find that ablating ASK neurons or reducing their activity using halorhodopsin does not disrupt either $O_2$-evoked changes in locomotory activity, or aggregation behavior. These data suggest ASK is not required for either of these responses, although we cannot exclude that it acts redundantly with other neurons to promote these behaviors.

The ASK ablation result contrasts with both our own data (*Tremain, 2004*) and those of others (*Macosko et al., 2009*). Both these studies show that *npr-1* animals defective in the cGMP gated ion channel TAX-4 fail to aggregate or to move rapidly in normoxia, and that these behaviors can be restored by expressing TAX-4 in URX and ASK (*Tremain, 2004*; *Macosko et al., 2009*). Why does ablating ASK have different behavioral consequences from disrupting cGMP signaling in this neuron? One possibility is that removing *tax-4* reduces the basal excitability of ASK (and other TAX-4-expressing neurons) and shunts current from RMG, compromising its ability to direct $O_2$ responses. However, since *tax-4* mutations disrupt many sensory modalities (*de Bono and Maricq, 2005*), other explanations are also possible.

## OCR-2-expressing neurons facilitate RMG interneuron signaling

Disrupting the OCR-2 TRPV-like cation channel attenuates the switch in locomotory state evoked in *npr-1* animals by 21% $O_2$. Restoring *ocr-2* to any one of the ASH, ADL, or ADF neurons restores $O_2$ modulation of locomotion to *ocr-2*; *npr-1* animals. Injecting current in ASH neurons using channelrhodopsin also restores strong modulation of locomotion by $O_2$ to *ocr-2*; *npr-1* mutants, consistent with *ocr-2* mutations chronically reducing ASH activity. The neuroanatomy suggests that ASH and ADL are gap-junctionally connected to RMG. Although a rise in $O_2$ evokes an increase in ASH $Ca^{2+}$, several findings argue against a simple 'hub-to-spoke' model in which ASH and ADL spokes are

necessary to relay RMG hub activity outwards to evoke behavioral responses. The $Ca^{2+}$ increase in ASH upon switch to 21% $O_2$ is very small. Ablating ASH and ADL, or expressing tetanus toxin in these neurons, does not disrupt the $O_2$ regulation of locomotion. Moreover, selective expression of *ocr-2* in ADF neurons, which are not gap-junctionally connected to RMG, can partially rescue the *ocr-2*; *npr-1* phenotype. Nevertheless, OCR-2 currents in ASH or ADL can influence the $O_2$-evoked changes in locomotion. How? One model is that tonic/chronic OCR-2 channel activity in ASH and ADL keeps these neurons depolarized and limits the current shunted locally from RMG to these neurons. Several observations support this possibility. First, expression of neuropeptide reporters in RMG is downregulated in *ocr-2*; *npr-1* animals to levels found in *npr-1* animals kept at 7% $O_2$. This suggests that loss of TRPV activity reduces neurosecretion from RMG. Second, the ability of *ocr-2* expression in ASH or ADL neurons to rescue the *ocr-2*; *npr-1* phenotype does not require neurotransmission, suggesting gap junctions are involved. Third, injecting current into ASH can restore to *ocr-2*; *npr-1* animals the $O_2$ regulated switch in high locomotory activity. Fourth, injecting current directly into RMG using channelrhodopsin can also rescue the *ocr-2*; *npr-1* phenotype, indicating that increasing RMG $Ca^{2+}$ is sufficient to bypass the lack of OCR-2-dependent activity. Unexpectedly, deleting *ocr-2* did not affect the steady state $O_2$-evoked $Ca^{2+}$ responses in the RMG cell body, suggesting that any effects on RMG $Ca^{2+}$ dynamics are local. The simplest model that explains our data is that OCR-2 signaling in ASH and ADL prevents shunting of current, or loss of a $Ca^{2+}$-dependent second messenger, from RMG to ASH and ADL through gap junctions. Membrane potential and $Ca^{2+}$ dynamics are not necessarily coupled, and a local current leak through gap junctions can be functionally important without necessarily altering $Ca^{2+}$ at the soma.

## RMG outputs

We show that none of the spoke neurons of the RMG 'hub and spoke' are individually necessary for RMG to alter the behavioral state in response to changes in $O_2$ in *npr-1* animals. This contrasts with the analysis of an anatomically similar circuit that mediates nose touch perception in *C. elegans* (*Chatzigeorgiou and Schafer, 2011*). In that circuit, OLQ and CEP mechanoreceptors that are coupled by gap junctions to each other and to the RIH hub interneuron act as coincidence detectors, pooling information through RIH, which in turn enables the high threshold FLP mechanoreceptors, also connected to RIH via gap junctions, to evoke a response to a gentle nose touch. In the RMG circuit, reducing RMG neurosecretory output by cell-selective expression of tetanus toxin or RNAi knockdown of the carboxypeptidase E *egl-21* strongly reduces transmission of the high $O_2$ information, suggesting that peptidergic release from RMG is a major output of the URX and RMG couple.

## AVB and AIY mediate some of the $O_2$-evoked behavior

Optogenetic experiments coupled with imaging experiments in freely moving animals provide insights into the information carried by different neurons in the $O_2$ circuit. RMG tracks $O_2$ levels, becoming more active at higher $O_2$ levels. Increased RMG activity drives faster forward movement and inhibits short reversals. However, RMG activity does not change when animals slow down or execute a reversal. Thus, RMG provides modulatory input but is not actually part of the circuit executing the behavior. Like RMG, AVB and AIY have, on average, increased $Ca^{2+}$ at 21% $O_2$. However, unlike RMG, these neurons do not show continuously high $Ca^{2+}$ levels. Instead, they show switches in $Ca^{2+}$ levels that correlate with reversal behavior. Most likely, the connectivity within the reversal modules (AIA, AIB, AIY, AIZ, AVA, and AVB) shapes a sequence of neuronal activation triggering/terminating the reversals. Modulation of the AVB and AIY interneurons by $O_2$ acts on top of that patterned activity to control overall locomotory pattern. AIA, AIB, and AVA neurons do not appear to be modulated by $O_2$ levels, and are involved in the reversal behavioral sequence irrespective of $O_2$ concentration (*Figure 9D–F*).

For behavioral effectiveness, each module state should enhance a specific behavior while suppressing incompatible behaviors. Our results suggest a direct role for AVB and AIY in both promoting high speed and suppressing reversals at high [$O_2$]. Such functions have been proposed for these neurons in other contexts (*Kawano et al., 2011*). Interestingly, AIY was also proposed to be involved in the control of lifespan and metabolism, something also modulated by $O_2$ (*Shen et al., 2010*). Potentially, information about $O_2$ concentration could flow from URX–RMG to AVB and AIY in part through neuropeptide secretion.

## General organismic state

Persistent changes in behavior and physiology require persistent changes in neural network activity. We identified several layers of neurons that display sustained changes in activity according to ambient $O_2$. However, the logic of the activity of different layers differs. In the $O_2$-sensing neurons, URX, AQR, and PQR, $Ca^{2+}$ sensors reported $O_2$ levels, as expected (*Busch et al., 2012*). In the first interneuronal layer, including the RMG hub interneuron and potentially the PVP interneurons, $Ca^{2+}$ levels also tracked $O_2$ concentration, although these neurons probably relay other information besides $O_2$ concentration, such as activity of the TRPV expressing neurons. Moreover, the activity level of RMG controls behavioral state rather than directly commanding a specific action: RMG activity remains high at 21% $O_2$ regardless of the animal's speed or direction of travel. By contrast, the second and third layers of interneurons modulated by $O_2$ appear to be directly involved in generating specific behaviors. $Ca^{2+}$ changes in these neurons anticipate or report the animal's behavior. The activity of some but not all of these neurons, including AVB and AIY, is modulated but not set by $O_2$ input. These downstream interneurons are probably not dedicated to a small subset of sensory inputs, as RMG or PVP might be, but instead simultaneously translate multiple streams of information into the appropriate behaviors.

Changing RMG activity is sufficient to evoke the behavioral states associated with high and low $O_2$. Downstream of URX–RMG we find widespread modulation of neurons and neuropeptide expression/ secretion, highlighting the complexity of even a simple contextual cue, $O_2$. Neuropeptide signaling appears to be key to generating the behavioral switch. Similar to motivated states, organismic states re-organize the salience of different sensory cues and change the physiology of the animal. It is tempting to propose that peptides whose release is modulated by $O_2$ re-organize sensory responses and modify the physiology of *C. elegans*. Studies of cross-modulation of $CO_2$ avoidance by the $O_2$ circuit support this scenario (*Carrillo et al., 2013*; *Kodama-Namba et al., 2013*). A similar example of how activation of a neuron and secretion of its associated peptides can coordinate behavioral and physiological responses is provided by mammalian tonic nociceptive neurons. In addition to amplifying the nociceptive signal in the spinal cord, the release of tachykinin and CGRP peptides from the nociceptive neurons alters gene expression in the surrounding tissues, producing neurogenic inflammation (*Carlton, 2014*).

Behavioral states and emotions are under intense investigation in mammalian systems, but the circuits engaged in implementing these states are only partially mapped. Changes in the global state of mammals are proposed to involve altered activities in multiple brain areas. Here, we outline how a global organismic state is encoded in a system that can be more easily circumscribed and comprehensively dissected.

# Materials and methods

## Strain and genetics

*C. elegans* was grown under standard conditions unless otherwise indicated (*Sulston and Hodgkin, 1988*). To cultivate animals in defined $O_2$ environments, we used a Coy $O_2$ control glove box (Coy, Grass Lake, Michigan, USA).

## RNA seq

Sample preparation: To prevent aggregation behavior, which could confound our comparisons (*Andersen et al., 2014*), we grew animals at very low density on a thin lawn of *Escherichia coli* OP50 spread to the edges of a 6 cm petri dish. To obtain a thin lawn, we grew bacteria on NGM containing only 5% of the regular amount of peptone. We picked five gravid hermaphrodites onto the lawn, let them lay eggs for 2–3 hr, and then removed them. The ~40 eggs were allowed to hatch and grow to adult worms at the indicated $O_2$ concentration. Animals from 10 such plates were rinsed off in M9 buffer and their RNA was extracted using RNeasy Mini Kits (Qiagen, Germany). cDNA libraries were prepared using the TruSeq Stranded mRNA Sample Prep Kit (Illumina) and sequenced on an Illumina HiSeq 2500 platform. We generated independent libraries of biological replicates as follows: *npr-1*, four libraries for 21% $O_2$ and six for 7% $O_2$; *gcy-35*; *npr-1*, seven libraries for 21% $O_2$ and seven for 7% $O_2$. Approximately 30 million 50 bp single-end reads were produced for each sample.

RNA-seq data analysis: Reads were output in FASTQ format and their quality assessed using FastQC v0.11.2 (http://www.bioinformatics.babraham.ac.uk/projects/fastqc/). A small portion (3–4%) of reads containing over-represented sequences (e.g., Illumina adapters) identified by FastQC were removed with Trimmomatic v0.30 (*Bolger et al., 2014*). The remaining reads were aligned to the *C. elegans* genome (WBcel235) with TopHat v2.0.13 (*Kim et al., 2013*). TopHat was run using default parameters with the following exceptions: coverage search was disabled using –no-coverage-search; library-type was changed to fr-firststrand; and the WBcel235 transcriptome annotations were provided via –transcriptome-index. In addition, since *C. elegans* has comparatively short introns (*Steijger et al., 2013*), –min-intron-length and –min-segment-intron were both reduced to 30. The aligned reads were then processed using the Cufflinks suite v2.2.1 (*Trapnell et al., 2012*) to assemble transcripts and ultimately compute differential gene expression between conditions. Transcripts were assembled for each sample with the Cufflinks tool (*Trapnell et al., 2010*), again lowering –min-intron-length, as well as –overlap-radius, to 30. The sample transcript assemblies were then merged with reference annotations (WBcel235) using Cuffmerge to generate a single, overall transcript assembly. The Cuffquant tool, using this merged assembly, was then used to compute gene and transcript abundances for each sample. Finally, sample abundances were integrated with the merged assembly by Cuffdiff (*Trapnell et al., 2013*) to test for differential expression between all condition pairs. All the Cufflinks tools, aside from Cuffmerge, were used with the –library-type parameter again set to fr-firststrand and both –frag-bias-correct (*Roberts et al., 2011*) and –multi-read-correct enabled. A gene was considered as differentially expressed between two conditions if the *q*-value (p-value after Benjamini-Hochberg correction for multiple testing) for the given comparison was <0.05.

The gene-level differential expression output from Cuffdiff was used to generate the presented excel files, which were annotated with WormBase IDs (WBcel235) and InterPro domains using custom Perl scripts.

## Molecular biology

DNA cloning was carried out using standard methods (*Green and Sambrook, 2012*). Promoters used in this work included: *gcy-32* (*Yu et al., 1997*), *sra-9* (*Troemel et al., 1995*), *pdf-1* (*Barrios et al., 2012*), *flp-8* (*Kim and Li, 2004*), *flp-21* short (*Kim and Li, 2004*), *flp-21* long (*Macosko et al., 2009*), *gpa-11* (*Jansen et al., 1999*), *srh-220* ADL (F47C12.5) (*Troemel et al., 1995*), *srh-142* ADF (T08G3.3) (*Troemel et al., 1995*), *ins-1* (*Tomioka et al., 2006*), *gcy-28* (*Tsunozaki et al., 2008*), *sra-11* (*Troemel et al., 1995*), *ttx-3* (AIY fragment) (*Wenick and Hobert, 2004*), and *ncs-1* (*Macosko et al., 2009*). Promoters were cloned into position 1 of the Multisite Gateway system (Invitrogen). Expression clones used in the course of this work are listed in supplemental data.

## Microfluidics

Microfluidic chambers for $Ca^{2+}$ imaging and behavioral assays were cast as described previously (*Busch et al., 2012*).

## Behavioral assays

For each assay, 20–25 adult hermaphrodites were transferred onto NGM plates seeded 12–14 hr earlier with 40 μl of *E. coli* OP50. To control $O_2$ levels experienced by the worms, we placed a 200 μm deep square PDMS chamber on top of the agar pad, with inlets connected to a PHD 2000 Infusion syringe pump (Harvard Apparatus) delivering humidified gas mixtures at a flow rate of 2.5 ml/min. We began pumping 5 min prior to the start of the recording to ensure that the initial environment was in a steady state. Movies were recorded using FlyCapture on a Leica M165FC dissecting microscope with a Point Grey Grasshopper camera running at two frames per second.

Movies were analyzed using Zentracker, a custom-written Matlab software (available at https://github.com/wormtracker/zentracker). Speed was calculated as instantaneous worm centroid displacement between successive frames. Frames in which the shape of the worm had an eccentricity value (ratio of the two axes of an ellipse with the same second moments) smaller than 0.8 or a compactness value (perimeter²/area) smaller than 30, combined with a solidity value (area/convex hull) greater than 0.575 were identified as omega-turns. To detect reversals, sharp (greater than 60°) changes in direction were first identified. Each potential behavior pattern that could be assigned to each track was then examined using a heuristic algorithm based on penalizing any occurrences of continuous movements lasting for more than 7.5 s in the same direction that do not correspond to

forward movements, any time an omega-turn is not followed by forward movement, and cases where a worm does not spend more time in forward than in backward movement. Using this information, directions of movement were determined for each worm in each frame according to the prospective behavioral pattern with the lowest overall penalty score.

## Optogenetic stimulation

ChR2 codon-optimized for *C. elegans* and C-terminally tagged with mCitrine, or halorhodopsin tagged with mCherry, were expressed from the promoters of interest. Worms were grown on plates pre-seeded with 200 μl of the *E. coli* OP50, with 15 μl of 5 mM all trans-Retinal (Sigma) dissolved in 100% ethanol added to the bacterial lawn prior to picking the worms onto the plates. Control worms were grown in parallel on seeded plates onto which 15 μl of 100% ethanol was added without the all trans-Retinal. Worms were preselected for fluorescence in the neurons of interest, and then assayed as described above. Light stimuli were delivered using a Leica EL6000 mercury lamp filtered for ChR2 or NpHR excitation using a Leica GFP2 or ET DSR filter, respectively. To avoid unwanted light-activation of the optogenetic actuators, we filtered the trans-illumination light using a 595 nm long-pass red optical cast plastic filter (Edmund Optics) for ChR2, and a 705 nm long-pass colored glass filter (Thorlabs) for NpHR.

## Neural imaging

We imaged immobilized animals as described previously (*Busch et al., 2012*). The experimental setup to image freely moving animals on an agar pad is shown in *Figure 8—figure supplement 1*. Emission light filtered for CFP and YFP wavelengths (460–495 nm and 525–580 nm, respectively), separated using the Tu-Cam system (Andor) with a 510 nm longpass, was relayed onto two identical Photometrics Cascade II 1024 EMCCD cameras running in frame-transfer modes with 60 ms exposure times. The imaged worm was kept in the field of view by moving the Prior stage manually using a joystick, with the acceleration rate set to its lowest value so as to disturb the worm as little as possible. To reconstruct the speed and trajectory of the worm, the actual stage position was continuously logged while acquiring image stacks using Micromanger (*Edelstein et al., 2010*) or custom-written software.

To deliver gas stimuli to the worm, we placed a square PDMS chamber on top of the agar pad, with inlets connected to a PHD 2000 Infusion syringe pump (Harvard Apparatus) running at a flow rate of 50 μl/s. An electronic valve system placed between the syringes and the PDMS chamber allowed us to switch between two different gas mixtures in a controlled manner at pre-specified frames. Using a spot Optode (PreSens), we monitored the $O_2$ concentration within the chamber while switching between mixtures containing 7% and 21% $O_2$, and found that $O_2$ levels could be switched reliably using this configuration (*Figure 8—figure supplement 1*).

We used Neuron Analyzer, a custom-written Matlab program, to analyze the resulting image stacks (available at https://github.com/neuronanalyser/neuronanalyser). In movies in which the neuron of interest provided the brightest signal in the field of view, we simply tracked the brightest points in each channel. Where this was not a viable option due to the complexity of the expression pattern, we used a semi-supervised heuristic tracking approach in which in each frame the area within a 20-by-20 pixel square centered around the location of the centroid of the region of interest (ROI) in the previous frame was examined for potential matches. Regions whose intensity differed by less than 1/3 from that of the ROI identified in the previous frame, and whose centroids were located within a 10-pixel radius of the ROI location in the previous frame, were identified as candidates for the ROI in the current frame. If no such candidates were detected, user input was requested in identifying the ROI in the current frame. If exactly one such candidate was detected, it was identified as the ROI in the current frame. If more than one such candidate was detected, match quality scores were calculated for each candidate based on their differences in intensity and centroid location from the ROI in the previous frame. If one candidate had a better quality score in both respects than all others, it was identified as the ROI in the current frame; user input was otherwise requested. In cases where the heuristic tracking method struggled to identify the ROI reliably in an automatic way, we used a user-supervised tracking method, in which a custom-sized moveable region was specified using the GUI, restricting the tracking of the brightest point within this regon of interest.

Once the centroid of the neuron of interest was identified, we calculated the mean of the 20 brightest pixels within a 10-pixel radius of the centroids for both channels independently, and subtracted from this the median pixel intensity of the respective channels as the background. We then corrected for the crosstalk by subtracting 0.6 from the ratio values.

To extract the behavior of the worm during the recording, first a more accurate measure of worm position was obtained based on not only the stage position, but also the location, within the field of view, of a successfully tracked neuron, effectively calculating the position, in stage coordinates, of the centroid of the ROI. The path was then smoothed using a moving average filter, and potential short gaps in the position data were identified, which may reflect the neuron temporarily leaving the field of view due to unexpected worm movements during the recordings. Gaps shorter than 30 frames were then filled using a spline interpolant, while gaps longer than this were excluded from the analysis.

In order to identify reversals, the direction of movement, within the coordinate system of the stage, was calculated for each frame using a central differences method, and changes in direction between successive frames were obtained. Intervals in which the worm moved slower than 10 µm/s were flagged as stationary behavior, and were excluded from further analysis. For the remaining frames, the relative changes in direction between successive frames were then examined, and periods containing changes no larger than 7.5° were classified as consistent movement. Consistent movement lasting longer than 90 s was flagged automatically as forward movements. For consistent intervals lasting shorter than this, user input was requested about the direction of the movement.

## Statistics

Statistics for both $Ca^{2+}$ imaging and behavioral assays used the Mann–Whitney U-test. Where we sought to compare steady state values, we chose time intervals where we expected the fluorescence ratios, or behavioral parameters, to have plateaued, that is, with a delay with respect to the timing of the switch in $O_2$ concentration or the presentation of blue or green light.

$Ca^{2+}$ imaging: when presenting time series over the duration of an experiment, n refers to the number of animals imaged. When presenting event-triggered averages (e.g., time relative to the onset of a reversal), n indicates the number of events. In this case, each animal was still only imaged once, but each animal could contribute to multiple events.

Behavior: for the intervals of interest, we extracted independent per-subject means deriving from worms flagged as continuously valid for at least 10 s during the interval. A worm was considered valid at a time point if it was not in contact with another animal, it was on the food lawn, and it was located at least half a worm-length from the border. Following these criteria, each worm was sampled at most once per interval; n indicates the minimum number of samples obtained per interval for the two intervals being compared.

## Acknowledgements

We thank Dr Yuichi Iino, Dr Chris Li, Dr P Sengupta, Dr R Shingai, the *C. elegans* Knockout Consortium, the National BioResource Project (Japan) and the *Caenorhabditis* Genetics Centre for strains, and de Bono and Schafer lab members for comments and advice. We are grateful to James Hadfield and Michelle Pugh at the Cambridge Institute Genomics Core for the RNA sequencing. ZS was supported by a Studentship from the Medical Research Council. MdB acknowledges support from the European Research Council (Advanced ERC Grant 269058).

## Additional information

### Funding

| Funder | Grant reference | Author |
| --- | --- | --- |
| Medical Research Council (MRC) | Studentship | Zoltan Soltesz |
| European Research Council (ERC) | 269058 | Mario de Bono |

The funders had no role in study design, data collection and interpretation, or the decision to submit the work for publication.

## Author contributions
PL, ZS, GMN, CC, FA-C, MB, Conception and design, Acquisition of data, Analysis and interpretation of data, Drafting or revising the article; EL, Acquisition of data, Analysis and interpretation of data, Drafting or revising the article

## Author ORCIDs
Geoffrey M Nelson, http://orcid.org/0000-0001-9825-4241

## Additional files

### Supplementary files
• Supplementary file 1. Comparison of RNA sequence data from *npr-1* young adults grown at 21% and 7% $O_2$.

• Supplementary file 2. Comparison of RNA sequence data from *gcy-35*; *npr-1* and *npr-1* young adults grown at 21% $O_2$.

• Supplementary file 3. Comparison of RNA sequence data from *gcy-35*; *npr-1* young adults grown at 21% and 7% $O_2$.

• Supplementary file 4. List of genes differentially expressed in *Supplementary file 1* (*npr-1* 21% $O_2$ vs *npr-1* 7% $O_2$) and *Supplementary file 2* (*gcy-35*; *npr-1* 21% $O_2$ vs *npr-1* 21% $O_2$) but not *Supplementary file 3* (*gcy-35*; *npr*-1 21% $O_2$ vs *gcy-35*; *npr-1* 7% $O_2$).

• Supplementary file 5. Lists of strains and constructs.

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
