## [Decision Letter]

Thank you for sending your work entitled “Decoding a neural circuit controlling global animal state in *C. elegans*” for consideration at *eLife*. Your article has been favorably evaluated by Catherine Dulac (Senior editor), Graeme Davis (Reviewing editor), and 2 reviewers (Scott Emmons and L René García), both of whom are expert in the research area of the manuscript.

The Reviewing editor and the reviewers discussed their comments before we reached this decision, and the Reviewing editor has assembled the following comments to help you prepare a revised submission.

Laurent et al. are reverse engineering circuit function in *C. elegans*. The circuit they are studying is one that results in multiple behavioral and physiological changes in response to elevated oxygen levels. Laurent et al. use the full range of experimental tools available in *C. elegans*, including cell ablation, Ca^2+^ imaging, optogenetic stimulation, cell-specific expression of transgenes to block chemical transmission and neuropeptide transmission or the cell-specific rescue of such functions in mutants, and automated behavioral analysis. They show that they can identify several layers of neurons in the pathway. Overall the data are convincing and clearly presented in the figures.

Both reviewers consider the topic interesting and consider the data well presented and supportive of your major conclusions. As you will see from the comments, both reviewers recommend significant re-writing of the text to make the manuscript accessible to the general audience of *eLife*. Both reviewers list a series of suggestions that you can use as a guide to improve accessibility of your text.

*Reviewer #1*:

1) The authors should choose one way to refer to *npr-1* “mutants.” This can be very confusing because the N2 allele is itself a “mutant.” The authors refer to the null allele in various places as *npr-1*, *npr-1* mutant, *npr-1* loss of function, *npr-1(null)*, *npr-1(ad609)*, and *npr-1(ky13)*. As *ky13* is specifically referred to as a null, what is *ad609*? In the second sentence of the Results they already refer to “*npr-1* mutants.” They should explain that by this they mean a null mutation (is that right?). They should specify how they intend to refer to the null allele, and then stick to that one form throughout the paper (except where they want to specifically indicate *ad609* vs *ky13*).

2) The first section of results on changes in gene expression is very confusing and needs rewriting for clarity. URX, AQR, and PQR neurons control “part, but not all, of this response.” What part? Where is the clear statement of the number of genes that change their expression between 7% and 21% O_2_ in *gcy-36*(+) but not *gcy-36*(–)? Figure 1 does not show this and therefore does not show what it says it does in its heading. It shows genes that change between *gcy-36*(+) vs *gcy-36*(–) at a single O_2_ concentration (21%).

3) Start the third section of Results by saying: “Among the three O_2_ sensing neurons, the URX neurons uniquely make gap junctions and reciprocal synaptic connections with the RMG interneurons.” Since in the forgoing URX has always been mentioned as one of a group of three sensory neurons, along with AQR and PQR, the reader makes a wrong assumption and puzzles, why doesn't RMG respond to AQR and PQR when URX is ablated?

4) For all their references to neural connectivity in the *C. elegans* wiring diagram, the authors should not only refer to the canonical White et al., paper but also to the data on the WormWiring.org website. The Emmons laboratory has reevaluated the original electron micrographs and has found some discrepancies.

5) The authors show that stimulating RMG can induce the high O_2_ behavioral response, so its activity is sufficient. But is it necessary? Where is the RMG ablation result? Has this been published already? Yes, I see, it is in Busch et al., Nature Neuroscience, 2012. This should be referred to, as well as their earlier data that the other two O_2_ sensing neurons signal via peptides to a different target. It would be helpful to make clear that the present paper focuses exclusively on the URX-RMG pathway.

*Reviewer #2*:

1) The meat of the study begins with Figure 2. As a reader that is not intimately acquainted with the O_2_-sensing circuitry, I found it helpful to draw out the circuitry and include where *ocr-2*, *npr-1*, *tax-2*, *flip-11*, etc. were expressed. As I was going through each of the experiments, I referred to my drawing to get my bearings on which part of the circuitry the authors were probing. I suggest that the authors provide a similar circuit roadmap as supplementary figure, to provide a reference for the reader to follow throughout the paper.

2) Beginning in Figure 2, as well as in the other figures, I was not sure what N refers to. For example in Figure 2 did not know if N referred to the number of animals? Or Did N refer to number of times that a 7%-21%-7% train was conducted on a limited number of animals, (if so how many animals?). This could be elaborated in the figure legend. Also in Figure 3, and elsewhere, what does N> (some number) mean? The graphs have SEMs, so there must be a discreet sample size.

3) In Figure D, E and F, (as well as in later experiments), the animals are not reported to contain the *lite-1* mutation, but in Figure 2, the strains did contain *lite-1*; is this just a typo? On the scopes in my lab, the 430 to 480 nM light will elicit an escape response in the worm (unless they are *lite-1*), if the light is too intense. You might want to mention in your Material and methods how you were able get around this. I noticed in Figure 5 (no retinal) that the light source on your set-up did not irritate the animals, so you might want to mention that early on.

4) You should clarify in the Material and methods what criteria you chose to pick the intervals (black bars and red bars) that you did your statistics.

5) In the subsection headed “ASK, ASH and ADL sensory neuron which have gap junctions with RMG are not required for the O_2_-evoked switch in behavioral state”, I was not sure whether the punchline was that these 3 neurons were not relevant for O_2_ sensation, or that these 3 neurons were redundant with each other for O_2_-responses. The latter seemed to be the case given the following section on *ocr-2* mutant phenotype, but stronger wording should clarify things.

6) In the subsection headed “Homeostatic re-setting of downstream O_2_ responsive circuits according to input”, the first sentence, “… alters circuitry by a homeostatic mechanism, such as…”: the use of the word homeostatic was not clear to me. My imagination (after pondering this for minutes upon minutes) started to come up with wild ideas of acclimation and desensitization after prolonged stimulation, with the ultimate equilibration to a default behavioral state. Then my imagination wanted to know what is that default behavioral state, and this distracting thinking is not what your paper is about. If I interpret your paper's point correctly, it is about how the regulation of immediate locomotor transitions are realized in a physical set of cell connections. So again, maybe more elaboration and keeping the reader's train-of-thought from digressing can help this section.

7) In the subsection “TRPV channel activity alters O_2_ circuit function upstream of NPR-1”, the last sentence of the first paragraph (“Instead, we suggest that TRPV dependent currents in ASH…”) needs more elaboration; there is more than one concept being stated in this sentence. This sentence puzzled me, and I am guessing (and probably wrongly), that the authors are suggesting that the ASH and ADL have the potential to leach away some excitatory factor from RMG via gap junctions; however, OCR-2 in ASH and ADL can attenuate this? Anyway clarification should make their concepts more obvious. Similarly, three sentences later, the statement, “as the tetanus toxin experiments suggested…”, also contained multiple concepts. This sentence also took me awhile to digest. Again more clarification should make this section easier to understand, and help put the next section in more context.

---

## [Author Response]

Reviewer #1:

*1) The authors should choose one way to refer to* npr-1 “*mutants.*” *This can be very confusing because the N2 allele is itself a* “*mutant.*” *The authors refer to the null allele in various places as* npr-1*,* npr-1 *mutant,* npr-1 *loss of function,* npr-1(null)*,* npr-1(ad609)*, and* npr-1(ky13)*. As* ky13 *is specifically referred to as a null, what is* ad609*? In the second sentence of the Results they already refer to* “npr-1 *mutants.*” *They should explain that by this they mean a null mutation (is that right?). They should specify how they intend to refer to the null allele, and then stick to that one form throughout the paper (except where they want to specifically indicate* ad609 *vs* ky13*).*

We agree that the nomenclature is confusing. The *npr-1(ad609)* behaves genetically like a null mutant. We have followed our reviewer’s advice: we state that we study *npr-1(null)* mutants and that we refer to these as “*npr-1*” throughout.

*2) The first section of results on changes in gene expression is very confusing and needs rewriting for clarity. URX, AQR, and PQR neurons control* “*part, but not all, of this response.*” *What part? Where is the clear statement of the number of genes that change their expression between 7% and 21% O*_*2*_
*in* gcy-36*(+) but not* gcy-36*(*–*)?*
Figure 1
*does not show this and therefore does not show what it says it does in its heading. It shows genes that change between* gcy-36*(+) vs* gcy-36*(*–*) at a single O*_*2*_
*concentration (21%).*

We have re-written this section entirely, and hope it is now clearer. We agreed with our reviewer that comparing gene expression in *gcy-35; npr-1* animals grown at both 21% and 7% O_2_ was a highly desirable control. As we had not analyzed *gcy-35; npr-1* expression at 7% O_2_ in our original microarray experiment, we repeated our study, this time using RNA sequencing, and included this condition.

Briefly, in our revised manuscript we use RNA Seq to compare gene expression of young adult *npr-1* and *gcy-35; npr-1* animals grown at 7% and 21% O_2_. In repeating the experiment we also took the opportunity to grow animals at very low density under conditions where we could see no aggregation. In this way we minimized changes in gene expression that are a consequence of aggregation behavior (see [1]).

3) Start the third section of Results by saying: “Among the three O_2_ sensing neurons, the URX neurons uniquely make gap junctions and reciprocal synaptic connections with the RMG interneurons.” Since in the forgoing URX has always been mentioned as one of a group of three sensory neurons, along with AQR and PQR, the reader makes a wrong assumption and puzzles, why doesn't RMG respond to AQR and PQR when URX is ablated?

We have made this change. We agree it clarifies the text.

*4) For all their references to neural connectivity in* the C. elegans *wiring diagram, the authors should not only refer to the canonical White et al., paper but also to the data on the*
*WormWiring.org*
*website. The Emmons laboratory has reevaluated the original electron micrographs and has found some discrepancies*.

We have added references to WormWiring.org throughout the manuscript.

5) The authors show that stimulating RMG can induce the high O_2_ behavioral response, so its activity is sufficient. But is it necessary? Where is the RMG ablation result? Has this been published already? Yes, I see, it is in Busch et al., Nature Neuroscience, 2012. This should be referred to, as well as their earlier data that the other two O_2_ sensing neurons signal via peptides to a different target. It would be helpful to make clear that the present paper focuses exclusively on the URX-RMG pathway.

We have made these changes.

Reviewer #2:

*1) The meat of the study begins with*
Figure 2*. As a reader that is not intimately acquainted with the O*_*2*_*-sensing circuitry, I found it helpful to draw out the circuitry and include where* ocr-2*,* npr-1*,* tax-2, flip-11*, etc. were expressed. As I was going through each of the experiments, I referred to my drawing to get my bearings on which part of the circuitry the authors were probing. I suggest that the authors provide a similar circuit roadmap as supplementary figure, to provide a reference for the reader to follow throughout the paper.*

We thank our reviewer for this useful suggestion. We have added a cartoon of the circuit in Figure 1—figure supplement 1. To keep the paper more manageable we have also removed Figure 1 and presented our gene expression data only in Supplementary data.

*2) Beginning in Figure 2, as well as in the other figures, I was not sure what N refers to. For example in Figure 2 did not know if N referred to the number of animals? Or Did N refer to number of times that a 7%-21%-7% train was conducted on a limited number of animals, (if so how many animals?). This could be elaborated in the figure legend. Also in Figure 3, and elsewhere, what does N> (some number) mean? The graphs have SEMs, so there must be a discreet sample size*.

For behavioral experiments, n refers to the number of independent worms in the time interval for which the statistic was calculated. The > sign (instead of an = sign) is to indicate that there are more worms on the plate than considered for the purposes of statistics. This is because some animals leave the field of view over the course of an experiment. Therefore, when comparing two intervals, one time point may have a smaller n-number than the other. In such situations we report the smaller n-number. In our revised manuscript we have explained this in the Methods, but removed the > sign as it confuses rather than illuminates.

For Ca^2+^ imaging each animal was imaged once, and n refers to the total number of animals studied. The exception is when we show event-triggered averages (e.g. time relative to reversal onset), in which case n refers to the number of events. In this case each animal was still only imaged once, but each animal could contribute to multiple events.

We say this in a section of our Methods dedicated to statistics.

*3) In Figure D, E and F, (as well as in later experiments), the animals are not reported to contain the lite-1 mutation, but in Figure 2, the strains did contain lite-1; is this just a typo? On the scopes in my lab, the 430 to 480 nM light will elicit an escape response in the worm (unless they are lite-1), if the light is too intense. You might want to mention in your Material and methods how you were able get around this. I noticed in Figure 5 (no retinal) that the light source on your set-up did not irritate the animals, so you might want to mention that early on*.

We now say that by using low light intensities we could evoke ChR2-dependent behavioral responses without irritating animals. Note that for most of our optogenetic experiments we used lower levels of blue light to activate ChR2 than used in most published work.

*4) You should clarify in the Material and methods what criteria you chose to pick the intervals (black bars and red bars) that you did your statistics*.

Done. For Ca^2+^ imaging we say: “As much as possible, we chose time intervals for statistical comparison when fluorescence ratios had stabilized. For behavioral experiments we avoid time intervals close to when we switch O_2_ concentration”.

*5) In the subsection headed “ASK, ASH and ADL sensory neuron which have gap junctions with RMG are not required for the O_2_-evoked switch in behavioral state”, I was not sure whether the punchline was that these 3 neurons were not relevant for O_2_ sensation, or that these 3 neurons were redundant with each other for O_2_-responses. The latter seemed to be the case given the following section on ocr-2 mutant phenotype, but stronger wording should clarify things*.

We have changed the wording to clarify this. However, as we have not ablated ASK, ASH and ADL together, we cannot explicitly state whether all three neurons are dispensable or not.

*6) In the subsection headed “Homeostatic re-setting of downstream O_2_ responsive circuits according to input”, the first sentence, “… alters circuitry by a homeostatic mechanism, such as…”: the use of the word homeostatic was not clear to me. My imagination (after pondering this for minutes upon minutes) started to come up with wild ideas of acclimation and desensitization after prolonged stimulation, with the ultimate equilibration to a default behavioral state. Then my imagination wanted to know what is that default behavioral state, and this distracting thinking is not what your paper is about. If I interpret your paper's point correctly, it is about how the regulation of immediate locomotor transitions are realized in a physical set of cell connections. So again, maybe more elaboration and keeping the reader's train-of-thought from digressing can help this section*.

We debated whether to take this section out or to keep it. On the one hand, keeping it allows us to explain why the speed of mutants kept at 7% O_2_ is higher than that of controls, an observation that initially puzzled us. On the other hand, as our reviewer highlighted, it distracts from our main message. Given the length of the paper we decided to remove it.

*7) In the subsection “TRPV channel activity alters O_2_ circuit function upstream of NPR-1”, the last sentence of the first paragraph (“Instead, we suggest that TRPV dependent currents in ASH…”) needs more elaboration; there is more than one concept being stated in this sentence. This sentence puzzled me, and I am guessing (and probably wrongly), that the authors are suggesting that the ASH and ADL have the potential to leach away some excitatory factor from RMG via gap junctions; however, OCR-2 in ASH and ADL can attenuate this? Anyway clarification should make their concepts more obvious. Similarly, three sentences later, the statement, “as the tetanus toxin experiments suggested…”, also contained multiple concepts. This sentence also took me awhile to digest. Again more clarification should make this section easier to understand, and help put the next section in more context*.

We agree these sentences are complex. We have tried to simplify them.